# Prion-Like Propagation Mechanisms in Tauopathies and Traumatic Brain Injury: Challenges and Prospects

**DOI:** 10.3390/biom10111487

**Published:** 2020-10-27

**Authors:** Hadeel Alyenbaawi, W. Ted Allison, Sue-Ann Mok

**Affiliations:** 1Centre for Prions & Protein Folding Disease, University of Alberta, Edmonton, AB T6G 2M8, Canada; alyenbaa@ualberta.ca (H.A.); ted.allison@ualberta.ca (W.T.A.); 2Department of Medical Genetics, University of Alberta, Edmonton, AB T6G 2H7, Canada; 3Department of Medical Laboratories, Majmaah University, Majmaah 11952, Saudi Arabia; 4Department of Biological Sciences, University of Alberta, Edmonton, AB T6G 2E9, Canada; 5Department of Biochemistry, University of Alberta, Edmonton, AB T6G 2H7, Canada

**Keywords:** proteinopathies, prionoid, dementia, transmission, seeding, concussion, clearance, tau genetics, strains

## Abstract

The accumulation of tau protein in the form of filamentous aggregates is a hallmark of many neurodegenerative diseases such as Alzheimer’s disease (AD) and chronic traumatic encephalopathy (CTE). These dementias share traumatic brain injury (TBI) as a prominent risk factor. Tau aggregates can transfer between cells and tissues in a “prion-like” manner, where they initiate the templated misfolding of normal tau molecules. This enables the spread of tau pathology to distinct parts of the brain. The evidence that tauopathies spread via prion-like mechanisms is considerable, but work detailing the mechanisms of spread has mostly used in vitro platforms that cannot fully reveal the tissue-level vectors or etiology of progression. We review these issues and then briefly use TBI and CTE as a case study to illustrate aspects of tauopathy that warrant further attention in vivo. These include seizures and sleep/wake disturbances, emphasizing the urgent need for improved animal models. Dissecting these mechanisms of tauopathy progression continues to provide fresh inspiration for the design of diagnostic and therapeutic approaches.

## 1. Introduction

Pathological tau aggregation is the defining characteristic for a subset of devastating neurodegenerative disorders collectively called tauopathies. In tauopathies, the microtubule-associated protein tau undergoes a plethora of changes at both the molecular and structural level, leading to the production of abnormal aggregates. These aggregates can take on the form of oligomers or larger filaments that assemble into neurofibrillary or glial fibrillary tangles [1,2,3]. Hyper-phosphorylated tau is a regular component of neurofibrillary tangles in Alzheimer’s disease (AD), and site-specific phosphorylation is a standard pathological marker of tauopathies [4,5,6]. Collective evidence linking tau aggregation to disease has prompted intense interest in the physiological and pathological roles of tau proteins. Recent studies demonstrate support for the prion-like hypothesis that tau protein pathology can be spread between tissues or brain regions [7]. Here, we provide a brief review of tau protein in healthy physiology and pathology (Section 2). We then evaluate recent evidence supporting the prion-like spreading of tau pathology, noting knowledge gaps and opportunities regarding traumatic brain injury (TBI) and chronic traumatic encephalopathy (CTE) (Section 3). Our discussion culminates in Section 4, which briefly considers TBI and CTE as a case study of the challenges and opportunities that emerge from considering their progression via prion-like mechanisms.

## 2. Tau Protein

This section provides a brief overview of tau protein, its isoforms generated from the *MAPT* gene, and their key roles in tauopathies. Human genetics and physiological roles for tau in healthy brains are described, as these set the stage for an appreciation of tau misfolding and its entanglement with tau post-translational modifications. The misfolding and oligomerization/fibrillization of tau are key amongst various biochemical/intracellular events that are considered prerequisite to the prion-like progression (i.e., non-cell-autonomous spreading) of tauopathy. These aspects of disease also are informative in a comparative framework, as they often provide correlative similarities and differences across the various tauopathies. We highlight knowledge gaps regarding these aspects of tau in TBI and CTE.

### 2.1. The MAPT Gene Encodes Various Isoforms of Tau Protein

Tau is a microtubule-associated protein that plays important roles in microtubule assembly and other microtubule-related functions [8,9]. It is encoded by the *MAPT* gene (Microtubule Associated Protein Tau), which is located on chromosome 17 (17q21.31) and comprises 16 exons, spanning ~150 kb [10]. Tau expression is detected in various mammalian tissues such as the heart, kidney, and skeletal muscles. Tau abundance is enriched in certain cell types of the central nervous system (CNS), predominantly in neurons [11,12]. It is also detected in oligodendrocytes throughout the brain [13,14]. In immature neurons, tau is expressed equally in both neurites and the cell body [15]. In mature neurons, tau protein is primarily localized in axons, where it shows a punctate distribution pattern [13,15,16,17]. Tau is also secreted from cells and is found in extracellular fluids such as the brain interstitial fluid (ISF) [18,19]. Tau expression in the brain displays intriguing regional differences, with both mRNA and protein levels reaching two-fold higher levels in the neocortex compared to the white matter and cerebellum [20,21]. This differential abundance in various brain regions may contribute to the differential vulnerability to tau pathology.

Six main isoforms of tau are found in the adult human brain. These are generated from the alternative splicing of exon 2, 3, and 10 (E2, E3, E10), as depicted in (Figure 1a). All six isoforms lack exon 4a, 6, and exon 8. Notably, the inclusion of exon 4a can result in the expression of a higher molecular weight (HMW) isoform of tau, called “Big tau”. Big tau is found mainly in the peripheral nervous system (PNS) or in CNS neurons that extend to the PNS [22]. Big tau and its splice variants are relatively unstudied in development, disease, and injury (readers are directed to a recent review by Fischer et al. (2020) for further reading [23]. Most studies related to tau, including those mentioned below, focus on the role of the six principal tau isoforms expressed in the CNS.

Tau isoforms can be distinguished via the alternative splicing of E2 and/or E3. This generates tau isoforms containing zero, one, or two amino-terminal inserts of 29 amino acid residues (known as 0N, 1N, and 2N, respectively; see Figure 1a). The six isoforms are also categorized as 4R or 3R based on the presence or absence of the second microtubule-binding repeat domain encoded by E10 (Figure 1a) [10,24]. 4R isoforms have a higher affinity for microtubules relative to 3R when assessed in vitro [25]. Recent models based on Cryo-EM structures of synthetic tau constructs bound to microtubules suggest that multiple microtubule repeat domains (see tau structure in Figure 1b) on a tau molecule can bind in tandem to the microtubule surface [26]. Combined with the evidence of identified binding sites in individual microtubule repeat domains [27], this model reveals a mechanism to explain the increased microtubule affinity of 4R tau isoforms. The functional consequences of the amino-terminal inserts of tau are not fully characterized. The inserts may modulate the subcellular localization of the protein [28]. There is evidence that 0N, 1N, and 2N isoforms may differ in their participating protein interactions, with 2N isoforms preferentially binding to proteins linked to neurological disease biomarkers [29]. However, increased 2N expression has also been linked to neuroprotection (see Section 2.3).

The expression of tau isoforms is developmentally regulated. In the human fetus, the 3R isoform is predominantly expressed [24]. In the healthy human adult brain, the ratio of 3R and 4R isoform expression is approximately equal with the 2N, 1N, and 0N tau isoforms, which encompass ~9%, 54%, and 37% of the total tau protein fraction, respectively [30]. The splicing of the *MAPT* gene also displays some regional variation. This is exemplified by the smallest isoform (0N3R), which has a lower expression in the cerebellum compared to the temporal cortex [20]. The mis-regulation of alternative *MAPT* splicing appears to be important for tau dysfunction because the resultant imbalance in 3R vs. 4R tau isoforms is correlated with distinct tauopathies (see next subsection).

### 2.2. Tauopathies Are Diverse But Can Be Categorized in Various Useful Ways

The term “tauopathy” was first introduced in 1997 [38] and is often used to describe a heterogeneous group of incurable neurodegenerative diseases. Despite diverse clinical presentations, tauopathies share a common pathological hallmark: the progressive deposit of filamentous tau inclusions. Tau aggregates vary across the tauopathies (Table 1) with respect to (i) their isoform composition (3R vs. 4R); (ii) their accumulation in disparate cell types (e.g., neurons or astrocytes); (iii) their morphologies and ultrastructure (e.g., spherical Pick bodies vs. neuro-filamentous tangles in neurons; paired helical filaments vs. straight helical filaments); and (iv) their propensity to affect distinct brain regions [39]. Although the majority of tauopathies are classified as sporadic (i.e., the causation is stochastic or it remains to be identified), a significant cohort of familial cases has also been identified [40].

Tauopathies have been categorized in various ways, including being described as primary or secondary in nature. In primary tauopathies, the tau inclusions are the main pathological marker. In secondary tauopathies, other protein aggregates are also found, such as Amyloid-beta (Aβ) in AD (Table 1) [39,41]. Some sporadic primary tauopathies are currently referred to as frontotemporal lobar degeneration-tau (FTLD-tau) based on the latest tauopathy classification scheme [39,42]. These include Pick’s disease (PiD), progressive supranuclear palsy (PSP), corticobasal degeneration (CBD), globular glial tauopathy (GGT), and argyrophilic grain disease (AGD) [39]. Familial tauopathies with *MAPT* mutations are often classified as FTDP-17 (familial frontotemporal dementia with Parkinsonism linked to chromosome-17) [40,43]. This designation has been problematic, since mutations in progranulin, which is also located on choromosome 17, cause tau-negative FTLD as well [44]. Therefore, we support a recent recommendation by Forrest et al., 2018, to denote FTLD associated with *MAPT* mutations as familial forms of FTLD-tau [45]. Their proposal is supported by the reanalysis of neuropathology data for a cohort of *MAPT* mutation cases that concluded each case could be readily classified into one of the sporadic primary tauopathy subtypes. Thus, the pathological mechanisms underlying sporadic and familial FLTD-tau may be more similar than previously considered.

CTE is categorized into the spectrum of tauopathies, as viewed in Table 1, as exhibiting a mixture of characteristics. The tau pathology in CTE is detected in both neurons and glia, along with other primary tauopathies, but displays a pattern of mixed tau isoform pathology most akin to that of Alzheimer’s Disease. It is worth considering that this categorization is relatively crude, insomuch that subtypes of the various disease may exist (e.g., sporadic vs. familial early onset AD). Moreover, CTE has a high degree of complexity, rooted in its variable onset associated with variation in the neurotrauma. This might include variation in TBI intensity, frequency, or mode or the age of the patient during exposure(s) (detailed in Section 4.3). Regardless, a pattern of similarities emerges, and it may be relevant that TBI is considered a risk factor most prominently for AD and CTE.

### 2.3. Human Genetics of MAPT Implicates Tau in Various Dementias

Knowledge from the disease-associated genetics of *MAPT*, which encodes tau protein, assisted in defining the potential molecular mechanisms linking tau dysfunction with neurodegeneration in tauopathies [60,61,62,63]. Tau gene expression and functions are altered by several genomic changes such as mutations and *MAPT* haplotypes.

Two main tau haplotypes, denoted as H1 and H2, occur due to an ancient inversion of a 900kb region that is centered around the *MAPT* gene [64,65]. The H1 haplotype is more common and diverse relative to the H2 haplotype. The H2 haplotype is found predominantly in persons with European ancestry or Caucasians [65,66]. With respect to the *MAPT* gene, H1 and H2 haplotypes are defined by several single nucleotide polymorphisms (SNPs) and a 238 bp deletion in intron 9 for H2 [64,67]. Genetic studies have highlighted the significance of haplotype-specific polymorphism in tauopathies. The H1 haplotype has an association with an increased risk of PSP and CBD, while H2 has a strong negative association with these diseases [61,64,68,69]. Additionally, H1 haplotype-specific variation in the MAPT 3′ UTR is associated with tangle-only dementia [70]. Since the two haplotypes do not alter the tau protein coding sequence, it has been suggested that the pathogenic effects observed with a particular haplotype may result from changes in splicing or transcription [61,71,72]. Indeed, allele-specific gene expression analysis has shown that the H2 haplotype produces a two-fold increase in *MAPT* transcripts that contain exons 2 and 3 in the gray matter area, which may indicate that 2N tau isoforms contribute to protection against neurodegeneration [71]. The H1 haplotype has been linked to a 40% increase in 4R tau isoform transcripts, consistent with 4R tau pathology and the affected regions observed in PSP [73].

*MAPT* mutations account for approximately 5% of familial FTLD cases. At least 59 pathogenic tau mutations had been linked to FTLD [74], and the list is expected to expand (see OMIM entry 157140). *MAPT* mutations are located in exons and intron regions primarily within the protein’s repeat domain region (Figure 1b) [75]. Mutations found in the exon regions can be missense, deletion, or silent in nature. Several altered properties have been attributed to mutant tau. These include affecting its microtubule binding properties; increasing its propensity to aggregate or be phosphorylated; and/or altering the splicing of *MAPT* pre-mRNA, leading to an imbalance in the ratio of tau isoforms (for recent in-depth reviews, see [39,74]). For example, similar to the potential link between haplotypes, splicing, and disease, several mutations in E10 and adjacent introns have been shown to alter the mRNA splicing of E10, leading to changes in 4R:3R tau isoform ratios [74,76,77] with abundant tau inclusions in neuronal cells. Multiple *MAPT* mutations have been shown to increase the propensity of tau protein to seed aggregation compared to WT, resulting in the development of tau pathology in animal models [78,79,80,81,82]. Collectively, multiple *MAPT* genotype-phenotype correlations provide strong support for the hypothesis that tau dysfunction plays a causal role in tauopathies.

Approaches to appreciate CTE and TBI outcomes using human genetics face several unique challenges compared to other tauopathies. In particular, genotype-phenotype correlations such as genome-wide association studies (GWAS) are made complex because neurotrauma is a substantial prominent risk factor that adds variability within the datasets. On the other hand, GWAS studies have been fairly consistent in identifying some loci such as *APOE* [83,84,85], and variants at other loci such as *TMEM106* have been given tentative support as being protective in CTE [86,87]. Thus, the power of human genetics to assess CTE is limited but not negligible. Tau protein is a potential biomarker for TBI or CTE, being increased in abundance in CSF following neurotrauma (see Section 3.6 and Section 4.1). *MAPT* variants and haplotypes had received tentative support as a risk factor in CTE; however, subsequent studies have failed to confirm that support [87]. Considering that a role for *MAPT* genetic variation in TBI outcomes and CTE would be exactly consistent with its role in the other tauopathies described above, we agree with the recent conclusion that further work is warranted in this area [88,89].

### 2.4. Physiological Functions of Tau

Tau is a multifunctional protein that can interact with microtubules (MTs) to regulate MT assembly, dynamic behaviors, and spatial organization [8,90,91,92]. Initial biochemical and cell studies supported a general role for tau in stabilizing MTs [27,93,94]. However, a recent study found that, in neurons, tau is enriched in the labile domains of axonal MTs, where it supports the increased length of these specific dynamic MT assemblies [95]. Tau protein also regulates axonal transport by influencing the functions of motor proteins such as dynein and kinesin, which transport cargo towards the cell body and axons termini, respectively [96,97]. Furthermore, several pieces of circumstantial evidence highlight that tau can regulate synaptic functions [16,98,99,100,101]. More recent studies implicate tau in the cellular processes of maintaining structures of axon initiation segments [102], protein translation [103,104], and response to DNA damage via effects on p53 [105]. Small amounts of nuclear tau detected in both neuronal and non-dividing cells may play a vital role in maintaining the stability of DNA and heterochromatin and regulating the degradation of cytoplasmic and nuclear RNA [106,107]. In oligodendrocytes, tau expression is directly linked to the myelination function of these cells [108,109]. The degree to which the perturbation of tau physiological functions may contribute to the pathogenesis of tauopathies is still unknown. However, several disease-associated mutations have been demonstrated to disrupt proposed tau functions such as microtubule binding and assembly or maintaining the axon initiation segment [30,74,102]. Future work might consider the engagement of tau’s physiological functions in response to TBI, and whether the misfolding of tau hinders effective responses to neurotrauma and subsequent CTE. This framework could offer an insightful complement to the mainstream priority of understanding the toxicity mechanisms that appear to be associated with the misfolding of tau across all tauopathies.

### 2.5. Tau Domains and PTMs in Native and Aggregated Tau Structures

Tau protein is comprised of two main major domains, which are defined based on their interactions with microtubules and their amino acid characteristics (see Figure 1b). A proline-rich region separates the *C*-terminal microtubule assembly domain and the *N*-terminal projection domain. The microtubule assembly domain contains a repetitive sub-domain (R1–R4) that contributes to binding to the microtubules and tau protein aggregation in disease [35,36]. The N-terminal projection domain is proposed to regulate tau secretion [110], plasma membrane binding, and synaptic protein function [111]. The proline-rich domain consists of several PXXP motifs that serve as binding sites for proline-directed kinases. The most well-studied kinases with respect to tau phosphorylation include Fyn [112], Glycogen Synthase Kinase-3 (GSK3) [113], and Cyclin-dependent kinase-5 (Cdk5) [114]. Functionally, the proline-rich region may support the microtubule binding of tau [115,116,117].

Tau is subject to complex and numerous post-translation modifications (PTMs). The PTMs implicated in physiology and/or pathology include phosphorylation, acetylation, *O*-glycosylation, *O*-GlcNAcylation, SUMOylation, prolyl-isomerization, ubiquitination, nitration, methylation, and protease cleavage [32,118,119,120,121,122,123,124]. An in-depth discussion of all tau PTMs is beyond the scope of this review, and readers are directed to further reading [124,125,126]. We will briefly discuss tau phosphorylation here (and in the subsequent section) to illustrate the complexity of tau PTMs. A total of 85 potential phosphorylation sites have been identified on tau [127,128,129,130], and the sites are distributed across the length of the tau sequence. The altered phosphorylation of tau has been linked to several of its physiological functions, including microtubule binding, axonal transport, and organelle delivery to the somatodendritic compartment [131,132,133]. The altered and increased phosphorylation (hyper-phosphorylation) of tau is associated with AD and other tauopathies [2,4,134,135]. Antibodies such as AT8 that recognize specific p-tau sequences have become standard pathology markers for tauopathies [136]. Whether or not tau phosphorylation promotes pathology such as tau aggregation will be discussed in the next section. Although a variety of kinases and phosphatases have been linked to the regulation of the tau phosphorylation levels [131,133,137,138], linking them to specific tau phosphorylation sites or tau functions has remained technically challenging.

Tau is a highly water-soluble (hydrophilic) protein under normal conditions due to both acidic and basic segments along the length of the protein [139,140]. Tau is classified as an intrinsically disordered protein that only transiently adopts secondary structure or global conformations [34]. For instance, data obtained from the analysis of global folding of the tau molecule in solution using electron paramagnetic resonance (EPR) and fluorescence resonance energy transfer (FRET) indicated that tau has a preference to change its conformation to “a paper clip-like” form in which the *C*-terminal end folds over the repeat domain and approaches the *N*-terminal end (Figure 2a) [141]. Tau appears to maintain its overall flexible, unstructured nature when interacting with other proteins or microtubules [26,93,142].

The aggregation process of tau is thought to be similar to that of other amyloid proteins, in which monomers self-associate and transiently sample conformations as small multimeric species known as oligomers [143]. A subset of oligomer conformations are thought to favor a transition to fibril structures in which the monomers become linearly stacked. These fibril structures are capable of rapid growth through the addition of monomers to the free ends. The ability of tau to undergo aggregation is dependent on the presence of two hexapeptide motifs called PHF6* (^275^VQIINK^280^) and PHF6 (^306^VQIVYK^311^), located at the start of R2 and R3 of the microtubule-binding repeat regions, respectively (Figure 1b) [35,144]. Early tau aggregation species known as oligomers have been ascribed to species as small as dimers and as large as 40-monomer units [145,146,147,148,149]. Tau oligomers are sometimes described as intermediates in the formation of fibrillar structures [149,150,151], but the challenges with isolating and maintaining oligomer species make it difficult to verify the specific precursor forms that lead to tau fibril formation. There are no high-resolution structural data for tau oligomers; however, several oligomer-specific antibodies have been developed, suggesting that oligomers possess distinct structures not shared by monomeric or fibrillar forms of tau [146,150,152]

The core of PHFs consists of a β-sheet structure which is characteristic of many amyloid-like structures [35,36]. Amino acid residues within the repeat domain of tau molecules form the structured core when tau protein aggregates into fibrils termed paired helical filaments (PHFs), whereas the remaining N-terminal and C-terminal tau residues form a “fuzzy coat” that surrounds the core of the filaments (Figure 2b) [36,153]. The imaging of the “fuzzy coat” is challenging due to its high flexibility, but it has been described as a two-layered “polyelectrolyte brush”, a structure that may be responsible for the stabilization of tau filaments [154]. Interestingly, recent advances in the analysis of tau filament structures from AD, Pick’s disease, CBD, and CTE samples using cryogenic electron microscopy denoted the ability of tau protein to adopt distinct folds and disease-specific conformations that may contribute to the neuropathological diversity observed in tauopathies [47,49,57,58]. The distinct conformations found in different tauopathies may be relevant to the concept that tau is prion-like, and this will be discussed in Section 3.3.

### 2.6. PTMs Correlate with Accumulation of Tau Aggregates: Cause and Effect

There are multiple biochemical aspects of tau that have been proposed to promote or potentiate tau aggregation, some of which are mentioned in the subsequent section. Here, we give an overview of why the post-translational modifications (PTMs) of tau are often touted to be one of the main drivers of tau aggregation in tauopathies, with a large focus on the incidence of tau hyper-phosphorylation [2,4,134,135]. Caution is warranted in dissecting whether these PTMs cause the accumulation of tau aggregates (and toxicity) and/or are a downstream consequence of tau aggregation. Dissecting this becomes increasingly difficult in vivo, though this will be an important challenge for the field as it seeks to understand the etiology of TBI and subsequent CTE.

For AD, the tau phosphorylation levels increase from 2 to 8 sites per tau molecule when compared to non-demented controls [6,155]. Follow up studies proposed that the detected levels of tau phosphorylation are likely underestimated, since post-mortem handling procedures may allow the significant de-phosphorylation of the sample to occur prior to analysis [32,156,157,158,159]. Several specific phosphorylation sites have been associated with tauopathies, particularly AD via immunohistochemical and, more recently, quantitative mass spectrometry techniques [128,136]. In some cases, the phosphorylation of a given site or a subset of sites on tau can even be shown to coincide with the individual Braak stages of AD progression [160,161]. Imbalances of kinase and phosphatase activities likely contribute to the detected changes in tau phosphorylation, but much work remains to sort out the complex contributions leading to specific patterns linked to disease [162]. In vitro studies using kinase p-tau or introduced phosphomimetic mutations have provided some clues as to how tau phosphorylation may promote aggregation [163,164,165,166]. Increasing the phosphorylation of tau at multiple or individual sites within the proline-rich and microtubule binding regions of tau has been shown to decrease its interaction with a major protein partner in the cell, microtubules [163,165]. Since microtubule binding involves the aggregation motif sequences within tau, a loss of binding may lead to the increased availability of these motifs to interact with other nearby, unbound, tau molecules.

Modulating tau phosphorylation has also been shown to potentiate its aggregation in vitro [163,164,165,166,167]. One hypothesis is that phosphorylation increases the overall negative charge of the tau molecule, which promotes aggregation. Interestingly, the location of individually phosphorylated sites modulates the kinetics of tau aggregation. However, combining phosphorylation sites did not always produce additive effects, which reveals some of the issues with deciphering the aggregation code of the complex tau phosphorylation patterns occurring in vivo [165]. Indeed, biopsy-derived human tau and transgenic mice studies have shown that most of the phosphorylation sites in PHF can also be phosphorylated in healthy brains [129,168]. At least some phosphorylation may occur prior to aggregation, as evidenced by a small subset of phosphorylation sites (e.g., Ser396 and Ser404) in the tightly packed core region of tau fibrils [169].

Moreover, in vivo data still only supports a correlative (not causative) role for tau phosphorylation in triggering initial aggregation processes or the prion-like spreading of aggregates between cells (discussed in Section 3.1). Several tau phosphorylation events are only linked to late stages of tauopathies, such as AD, well after NFTs have begun to accumulate [160,161,170]. An alternative possibility is that some (or most) disease-associated tau phosphorylation occurs post-aggregation. Likewise, it may be that phosphorylation (at some or most sites) is not a significant contributing factor to initiating tau aggregation events in disease. For example, the vast majority of tau phosphorylation sites lie in the fuzzy coat surrounding the characterized tau fibril core regions in tauopathies [49,57,58,137,171]. The disease-associated phosphorylation status of specific sites may simply be dictated by their accessibility to kinase and phosphatase actions within fibril structures [172]. Tau phosphorylation may also be pro-pathogenic via mechanisms independent of the phosphorylation’s potential role in tau aggregation. For instance, tau phosphorylation at Tyr18 enhances its promotion of excitotoxicity in neurons [173].

Other types of PTMs on tau may also promote tauopathy. Tau acetylation at residues Lys274 and Lys280 has been detected in the tau PHFs from multiple tauopathies [119,174]. Lys280 acetylation has been shown to enhance tau aggregation and impair the microtubule binding of tau in vitro, similar to phosphorylation effects described previously [175,176]. Tau acetylation may also promote the protein’s aggregation indirectly via multiple mechanisms including preventing its degradation, potentially by directly competing with ubiquitination processes for the modification of lysines [177,178]. Another tau PTM, the O-linked N-acetyl-d-glucosamine (O-GlcNac) modification of tau, is decreased in AD and other neurodegenerative diseases, suggesting that this modification has protective properties. O-GlcNacylation directly inhibits tau aggregation propensity in vitro and in vivo [179,180,181]. Moreover, O-GlcNac modification can directly compete with tau phosphorylation at the Ser and Thr sites in cells, suggesting that O-GlcNacylation may prevent at least a subset of the tau hyper-phosphorylation events linked to disease [182]. Each of these PTMs provide opportunity for further work in TBI and CTE, both as biomarkers and as tools to provide insights into etiology. It will be critical to consider their impacts in vivo and dissect the roles of various PTMs as being a cause or effect of aggregation (or toxicity), as described in the previous paragraph.

### 2.7. Other Biochemical Aspects That Impact Accumulation of Tau Aggregates

Efforts to appreciate what promotes tau aggregates, or what hinders their degradation and clearance, have typically required the power of a reductionist view and thus a deep appreciation of biochemical events. This subsection briefly introduces some informative mutations and isoforms of tau, some aspects of clearance, and the toxicity of tau aggregates. Further work is warranted in translating this knowledge to in vivo contexts and assessing its applicability to TBI and CTE.

Assessing impacts of these aspects in vivo is a key priority because in vitro and biochemical results can assign the incorrect consequence of tau phosphorylation. For example, it is instructive to consider that tau mutations causing FTLD-tau may promote aggregation and disease (discussed in Section 2.6) through multiple mechanisms. Mutations such as P301L and P301S have been well-studied for their ability to strongly enhance the intrinsic propensity of tau to aggregate. These mutants have been used to create multiple animal models of tauopathy [80,81], creating a critical bridge between biochemical analyses and in vivo outcomes. Tau P301L mutants, studied in isolation via NMR, may have a propensity for aggregation via an increased tendency of tau monomers to adopt more β-structure character [183]. An alternative mechanism suggests the P301L mutation disfavors a tau conformation that normally shields tau’s PHF6 aggregation motif, as revealed by crosslinking and synthetic tau peptide studies [184]. Considered instead in a cellular context, a subset of tau mutations including P301L has also been shown to more readily overcome the protective anti-aggregation effects of molecular chaperones [185]. Chaperones are the primary cellular factors that recognize and triage misfolded proteins in the cell, and further consideration of their roles in tau aggregation and degradation is warranted [185]. Interestingly, multiple tau mutations (V337M, R406W) create NFT pathology when introduced in mice [186,187], despite showing little or no ability to enhance the tau aggregation kinetics in vitro [82,188]. Models such as these could offer more direct insight into the cellular factors contributing to tau aggregation that may also be driving sporadic FTLD-tau.

In vitro studies have also suggested that several polyanionic molecules such as such as heparin, fatty acids, nucleic acid, and polyphosphates can trigger the formation of tau fibrils and filaments [189,190,191,192,193]. However, whether such molecules promote aggregation remains to be confirmed in vivo. A comparison of fibrils assembled using heparin vs. those isolated from AD and Pick’s Disease cases revealed differences in the conformations of the packed core regions [171]. This could indicate per se that heparin is not the sole trigger for tau aggregation in diseases such as AD, but it does not rule out the possibility that cellular cofactors have an important role in initiating aggregation in tauopathies. Tau fibrils from CTE and CBD patients have already been demonstrated to have unidentified densities that are postulated to be aggregating co-factors [57,171]. In this context, the complexity of polyanionic molecules is worth acknowledging. Intuitively, these substrates of tau misfolding must vary with time; differ based on the cellular compartment; and be distinct between neuronal, glial, or other cell types. Moreover, TBI and subsequent neuroinflammation are expected to alter this milieu. Future work might consider how this complexity promotes or retards prion-like tauopathy progression.

It has already been mentioned that 4R:3R isoform imbalance is linked to tauopathies via *MAPT* haplotypes and tau familial mutations. However, the mechanism underlying this has still not been elucidated. Tau 4R isoforms have an increased propensity to aggregate over 3R isoforms [194], likely due to the presence of an extra aggregation motif. Tau 3R isoforms have also been demonstrated to inhibit the aggregation of 4R tau when placed in the same aggregation reaction [194]. A mouse model that allows for the manipulation of 4R:3R tau ratios does support that increasing the proportion of 4R tau leads to more tau aggregate pathology [195]. These data may provide a mechanism for tau pathology in tauopathies linked to increased 4R tau. However, they do not lend insight into the cause of tauopathies featuring aggregates composed of 3R tau, such as Pick’s Disease.

It is reasonable to speculate that failure to clear or degrade tau or PTM-modified tau could lead to increased intracellular tau concentrations that facilitate aggregation [196]. The mechanisms involved in tau clearance may depend on the location of tau species (extracellular vs. intracellular tau) [197], type of tau molecules (truncated or modified via mutations or post-translational modification) [196], and other factors such as sleep and wake cycles [198]. Various studies have supported the clearance of tau in cells via two major degradation pathways, the proteasome system and the autophagy system [196,199,200]. It is known that PTMs and tau mutations can alter the clearance of tau via these systems. For example, introducing P301L or phosphomimetic mutations into tau impairs its degradation by the chaperone-mediated autophagy pathway [201]. In another example, the caspase cleavage of tau at Asp421 drives aggregation in in vitro studies and is linked to AD pathology [202]. Caspase cleavage at Asp421 of tau recruits the ubiquitin ligase CHIP to tau’s neo-C-terminus [203]. The decreased CHIP levels observed in AD may provide a mechanism for the build-up of caspase-cleaved tau also observed in AD pathology. The tau secreted from cells can be cleared by the glymphatic system [204,205], a glial-based waste clearance system to remove soluble proteins and metabolites from the CNS. Interestingly, the glymphatic clearance system predominantly operates during sleep, raising the possibility that major lifestyle components could contribute to disease risk. Recent work has shown that the incidence of dementia among shift and night workers seems to be modestly associated [206]. Chronic sleep disturbances have been shown to cause an early elevation of p-tau in P301S mice, leading to the progression of tauopathy including NFTs over time [207]. Furthermore, several tauopathies contribute themselves to disrupting sleep/wake cycles, setting the stage for a feed-forward mechanism promoting tau pathology.

Tau aggregates can exert toxicity from gain of function, such as interfering with cell signaling pathways or other intercellular functions [32]. Although NFTs are the histopathological marker for many tauopathies, it is not understood whether filamentous tau assemblies are accountable for the cytotoxicity observed in these diseases or if they are a productive response to cellular stress or aging [208,209]. Recent studies seem to support the latter, because NFTs can be detected in the brain of some aged people with no noticeable cognitive deficits; moreover, neurons with NFTs have been shown to be functionally intact in vivo [209,210]. Additionally, neuronal loss was observed before the formation of NFTs in a *Drosophila* model of tauopathy [211]. Finally, the suppression of disease–associated mutant tau expression in repressible transgenic mice led to recovery of memory functions and decreased neuronal loss, despite the continuation of NFT accumulation, arguing that NFTs alone may not be sufficient to promote cognitive deficits and neuronal death [80].

Lack of evidence for NFT toxicity leaves the possibility that smaller and “pre-tangle” assemblies of tau aggregates, specifically oligomers, may be the prime culprit behind toxicity [146,151]. For AD, tau oligomers have been detected at the earliest stages of the disease [146,147,149,212]. Tau oligomers have also been found in PSP samples [213]. Injection of tau oligomers into mice lead to measured synaptic and mitochondrial alterations neurotoxicity and memory deficits that were not observed in mice injected with tau fibrils [151,214]. In one study, the persistence of injected tau oligomers in mice could be reduced by injection with a tau oligomer antibody, and the treatment rescued associated cognitive defects [152]. Cell culture experiments demonstrate tau toxicity in some studies [150,152,215,216], but only subtle/no defects in other studies [217,218]. The conflicting results in these studies could be based on the cell model used, but another contributing factor is likely differences in the tau oligomer material used for these studies. Research groups utilize a wide variety of methods to isolate heterogeneous and unstable tau oligomer species: direct isolation from patient samples [151,152], the in vitro cross-seeding of recombinant tau with Aβ oligomers [214], and recombinant tau oligomers with or without stabilization via chemical cross-linking [215,216,217,218]. The label of “tau oligomers” in studies is not descriptive enough and does not accurately represent the diversity of tau oligomer species that can be generated across methods or how individual species might differentially influence tau aggregation and prion-like properties in vivo. Future work in the tau oligomer field could benefit from standardized isolation methods or molecular characterization that can be readily adopted by all groups, a movement that is also being suggested for other amyloidogenic proteins such as alpha-synuclein [219]. The complexity of the tau species generated following neurotrauma—e.g., TBI and subsequent CTE—will be an imposing challenge to any reductionist approach.

## 3. “Prion-Like” Properties of Tau

Tauopathies are considered to be prion-like diseases. The prion-like disease concept centers on appreciating certain overlapping characteristics between prion diseases and various progressive neurodegenerative diseases whose etiologies are propelled by protein misfolding. Post-TBI tauopathies including CTE appear to be prion-like, along with all other tauopathies. Appreciating this prion-like etiology offers potential for novel diagnostics, therapeutics, and prophylactics. Below, we first provide context (Section 3.1) via a brief description of classic prions, highlighting (i) the template misfolding of proteins and amyloidosis; (ii) the spreading of the misfolding in a non-cell-autonomous manner (promoting disease progression); and (iii) the existence of prion strains that can account for disparate disease outcomes, despite the shared root cause of misfolding one particular species of protein (e.g., misfolded prion protein causes various diseases; misfolded tau causes various distinct tauopathies). We then describe (Section 3.2) evidence supporting the notion that tauopathies have prion-like characters. We consider that strains of tau may account for distinct tauopathies, including CTE (Section 3.3). Finally, we describe the knowledge gaps surrounding how tau spreads between cells and between tissues (Section 3.4 through Section 3.8). The multitude of mechanisms and vectors of tauopathy spreading through the nervous system warrants careful consideration. Various tauopathies are considered here, as only a limited number of investigations have detailed prion-like aspects of tau regarding TBI and CTE. Appreciation of these is expected to be a fruitful source of novel diagnostics, therapeutics, and prophylactics in tauopathies such as TBI and CTE.

### 3.1. Prion-Like Diseases: Defining Characteristics

To best appreciate how tauopathy research has been influenced by the prion-like disease concept, a brief overview of classic prion diseases is presented.

Prion diseases are fatal neurodegenerative diseases linked to the pathological misfolding of the prion protein (PrP) [220]. The cellular form of prion protein (PrP^C^) is expressed mainly in the brain and is characterized as a ubiquitous glycoprotein largely localized to the plasma membrane in cells. Its C-terminal half is folded and features hydrophobic and alpha-helical structural elements. PrP^C^ can misfold and aggregate (“convert”) to generate highly stable, protease-resistant, fibril structures composed mainly of beta-sheets [221,222,223]. This aggregated state of the protein is classically designated as PrP scrapie (PrP^Sc^), with scrapie referencing a prion disease found in sheep [224,225]. Similar to tau, the cause of the initial misfolding of PrP^C^ molecules is an intense area of study with many proposed contributing factors, such as mutations or alterations in PTMs, metal binding, protein interactions, and cellular stress conditions. Perhaps the most striking and initially controversial property of PrP^Sc^ is that, when transferred via contaminated tissue or secretions to a susceptible naïve host, PrP^Sc^ alone is sufficient to trigger disease. The name prion is based off the original characterization of this transmissible property being described as a proteinaceous infectious particle [224]. A famous historical example of transmitted human prion disease, kuru, is attributed to the consumption of contaminated human tissue during tribal funeral rituals in Papua New Guinea. Humans are also susceptible to contracting prion disease from other species: variant Creutzfeld-Jakob Disease (vCJD) is the result of prion disease transmitted from cattle to humans through eating contaminated meat [226].

Decades of research have supported the concept that the progression of toxicity in the naïve host after PrP^Sc^ transfer is via the ability of the PrP^Sc^ aggregates to act as “seeds” that template the aggregation (also known as conversion) of a host cell’s monomeric PrP^C^, resulting in the formation of new seeds (for a review see [227]). When templated seeding or nucleation occurs in a cell, some seeds in the affected cells can escape and transfer to adjacent naïve cells or tissues by several potential mechanisms discussed in more detail in Section 3.3. The transfer of seeds and pathology from cell to cell or between brain regions is referred to as “spreading”. Strategies that block the templating or spreading of protein misfolding offer therapeutic potential.

Another defining characteristic of prion pathobiology is the concept of “strains”. The strain concept was developed based on data that PrP^Sc^ isolated from a given source (strain) and inoculated into naive hosts consistently resulted in a characteristic profile of clinical phenotypes, pathology, and PrP^Sc^ biochemical properties. The differences in the biochemical properties of individual strains such as resistance to protease digestion or denaturing agents [228] have led to the currently accepted view that strains can be separated based on structural differences in their PrP^sc^ aggregates that dictate seeding and pathogenic effects in vivo.

Evidence of PrP^Sc^ strains can be observed in the most common prion disease, sporadic Creutzfeld–Jakob Disease (sCJD). sCJD has a heterogeneous clinical and pathological presentation, and individual cases of sCJD can be further categorized into one of six subtypes [229]. Subtypes of sCJD correlate with the distinct heat and chemical denaturation resistance of PrP^Sc^ isolates, which is indicative of differences in PrP^Sc^ structures and consistent with the strain concept [228]. For sCJD, multiple strains can exist in different brain regions of the same patient, which may also help to account for the large phenotypic variability observed in this disease [230]. Strains linked to sCJD are readily distinguished from strains found in vCJD and familial forms of CJD [231,232].

Prior to the prion-like paradigm, cell-autonomous mechanisms combined with the concept of selective neuronal vulnerability were assumed to sufficiently explain the distinct patterns of neuronal loss associated with neurodegenerative diseases such as tauopathies [233,234]. The cell-autonomous mechanisms suggest that the same events, such as protein aggregation or the disrupted clearance of aggregates, can occur in brain cells independently. At the same time, the concept of neuronal vulnerability implies that subpopulations of neurons, varying depending on the disease, are intrinsically more vulnerable and are affected earlier in the disease than other cells. This vulnerability may be due to the cells’ gene expression profile and how it is modified during the ageing process [233]. Yet, the prion-like spreading of tau aggregation is also a plausible hypothesis by which to explain the observed pathological profiles of tauopathies. Prion-like spreading would fit with the typical progression of tau accumulation across connected brain regions in AD. It would also offer a mechanism by which the aggregation of the same protein, tau, can lead to the multiple clinical and pathological phenotypes defining the various tauopathies; each tauopathy could be caused by a distinct strain or strains. It is important to note that the cell autonomous/neuronal vulnerability hypothesis and the prion-like spreading hypothesis for tauopathies are not mutually exclusive. The formation of the initial seeds of a given strain could be largely dictated by cell autonomous events in neuronal populations vulnerable to that strain. Similarly, selective neuronal vulnerability may continue to influence the routes and rates of spreading for individual strains.

Initial evidence suggests that other amyloidogenic proteins linked to neurodegeneration might share the transmissible properties ascribed to prions and might thus be considered “prion-like”. We utilize the term prion-like to specifically describe the attributes of templating, strain, and cell to cell transmission shared by several amyloidogenic proteins. However, we also recognize that there is controversy in the field regarding the use of the term “prion-like” vs. other terms, such as “prionoid” or “Prions”, and we direct readers to a more targeted review on the subject [235]. The prion-like hypothesis has spurred a transformation of the research landscape for diseases such Alzheimer’s, Parkinson’s, and Huntington’s [59,236,237,238]. In the following sections, we provide an overview of data examining the prion-like properties of tau.

Considering the concepts of prion research summarized very briefly above, it is noteworthy that tauopathies are (i) initiated by protein misfolding that templates further tau misfolding; (ii) tau misfolding progresses into adjacent cells and brain regions in a fashion that may account for the progression of clinical symptoms; (iii) various tauopathies present with different neuropathologies (e.g., see Table 1) and biochemistries reminiscent of strains. We consider below and in Section 4 which of these prion-like aspects of tau might be applicable to TBI and CTE, and what areas require further research.

### 3.2. Evidence Supporting the Prion-Like Seeding and Spreading of Tau In Vitro and In Vivo

Alzheimer’s disease (AD) was amongst the first of the tauopathies to be investigated as a prion-like disease. In AD, the tau pathology progresses in a stereotypical pattern that is used to define specific stages of the disease [41]. At the earliest stage of AD, the accumulation of tau aggregates is detected in a small brain region called the transentorhinal cortex. In subsequent stages, tau pathology characteristically accumulates in anatomically or synaptically connected regions, consistent with the possibility of a cell-to-cell spreading mechanism for tau aggregation [41,239]. Recent radio imaging studies utilizing PET tracers to track tau aggregates (PHF-tau) propose that tau pathology spreads via neuronal networks in AD [240,241,242]. The transmission of tau aggregates through connected neural synapses may not be the only cellular mechanism that can mediate spreading in tauopathy. In PSP, tau pathology can be detected and spread in glial cells [242]. These data warrant the consideration of tau aggregate spreading mechanisms via other CNS connectivity routes, such as interstitial fluid or the glymphatic system.

The ability of tau seeds to initiate tau seeding was observed over two decades ago. In 1996, work conducted by Alonso and colleagues showed that hyper-phosphorylated tau isolated from AD can induce self-assembly when combined with recombinant tau [243]. Later in 2009, a similar observation was reported, in which the addition of external seeds of synthetic tau filaments led to tau seeding and accelerated aggregation into filaments in vitro [244]. The mentioned study also demonstrated the capacity of tau aggregates to transfer or be transferred between co-cultured cells. Around the same time, the prion-like mechanisms of tau were also evidenced in vivo by demonstrating that tau aggregate pathology could be transmitted to a naïve host [236]. Brain extract from a tauopathy mouse model (mutant P301S tau) with disease phenotypes including neurodegeneration and filament formation was injected into the brains of mice expressing human wild-type tau. These wild-type tau mice did not normally develop tau filaments or show neurodegeneration phenotypes. Remarkably, the injections of the P301S mouse brain extracts into these mice induced the seeding of wild-type tau into filaments and the spreading of pathology far from the injection site to disparate brain regions [236]. Afterward, various groups studied the prion-like propagation of tau using the same inoculation approach but with extracts from AD patients as well as synthetic tau filaments. Injections into both transgenic mice or wild-type mice produced similar results. Regardless of the extract source for pathogenic tau, endogenous tau was induced to aggregate, and tau aggregate spreading was evident in most cases [245,246,247]. The transmission of tau within the CNS was even observed after peripheral injections of tau aggregates [248]. While this approach supports the concept of tau propagation in a time-dependent matter, it provides only limited information on the detailed mechanism(s) of the propagation of tau and how it is released and taken up by other cells. Recently, the source of tau seeds used in the intracranial delivery experiments has caused some debate with regard to which source of tau aggregates (recombinant tau fibrils, or fibrils from patients or transgenic mice in the form of brain extract) may be more relevant to human tauopathies. This debate may be influenced by the revelation that the conformations of PHFs generated from recombinant tau proteins are different than the ones isolated from humans AD and other tauopathies such as PiD [171,247,249]. Additionally, the fibrils isolated from AD patients or transgenic mice have a better seeding capability than recombinant tau [249,250]. The fact that recombinant tau fibrils perform poorer in seeding assays should be exploited in comparative structural and biochemical studies to help the field to better understand the basis of tau strains and their associated seeding and spreading properties.

The spreading of tau pathology may involve both neuronal and non-neural cells of the CNS. The drug-induced depletion of microglia significantly reduced the spreading of tau pathology in a mutant tau (P301S) transgenic mouse model [251]. Oligodendrocytes have also been shown in vitro and in vivo to play a role in the spreading of the glial tau pathology of CBD and PSP tau strains, in a process that did not require neuronal tau [252]. Despite the evidence provided regarding the ability of tau to propagate in a prion-like manner, there is still lack of information about which tau species are involved in the seeding and spreading of the pathology, especially when considering variables such as the tau aggregate species (oligomeric vs. fibrillar) and the location of tau species (cellular tau vs. secreted tau). Mirbaha et al. proposed that trimers isolated from recombinant tau aggregates or AD tissues were the smallest size of tau aggregates that can induce seeding [145]. Follow-up studies from the same group put forth unexpected evidence that monomeric tau is capable of propagating strain information when used to seed aggregation in cells [253,254]. The heat-stable tau monomer conformations can either be categorized as “inert” with shielded aggregation motifs that do not aggregate spontaneously or as multiple distinct “seed competent” conformations which, in contrast, have exposed motifs and a higher tendency to self-seed. It is important to note that a similar study in primary neurons and mice using tau species isolated from brain extracts detected seeding via soluble oligomer tau species but not monomers [255]. Considering what was mentioned earlier regarding the use of recombinant tau alone (which is the case in some of the mentioned studies) and how relevant it is to tau in affected human cell types in the CNS, further validation is critical to support the present studies. This would include work addressing whether monomeric tau can efficiently seed aggregation in vivo and whether this seeding process possesses the characteristics of prion-like propagation.

### 3.3. Prion-Like Strains of Tau May Exist and Be Fertile Ground for Innovation

One of the characteristics of classical prions (PrP^Sc^) is their fascinating ability to propagate as “strains”, which refers to protein aggregates that have the same primary protein sequence yet assume distinct conformations that are stably maintained even when isolated from cells or organisms and then reintroduced to naïve cells or organisms [256]. Because of these conformational differences, prion strains have different biochemical properties, have different morphologies of protein aggregates, have different effects on disease onset and duration, and trigger different pathologies and clinical courses that affect distinct regions or specific tissues [257]. Each of these diverse characteristics of pathology can be passaged to naïve individuals with a high fidelity, despite there being no known variance in primary protein structure amongst the different strains, suggesting that information is encoded and transmitted via the folded elements of the protein. In tauopathies, the tau inclusions formed in each tauopathy differ in the cells they affect and their morphology, structures, and isoform composition (Table 1) [258,259]. In PiD, for instance, tau fibrils form round, interneuronal inclusions called Pick bodies that are composed only of 3R isoforms of tau and straight filaments [260,261,262]. In contrast, inclusions compromising 4R tau isoforms are commonly observed in the remaining FTLD-tau, including PSP, CBD, GGT, and AGD [75,263]. In AD, tangle-only dementia (TD) and CTE tangles are formed from both 3R and 4R isoforms [70,264,265]. Considering the diversity of tau inclusions formed in each tauopathy, a hypothesis was formed that, similarly to prions, different tau strains may exist that can induce distinct tauopathies. This notion initially was supported by observations from various in vitro and in vivo studies [247,266,267,268,269,270]. In the earliest studies, brain homogenate or intracerebral fluid from tauopathy patients (including PiD, AGD, PSP, and CBD) was injected into mice expressing the longest isoform of human tau (2N4R), which resulted in the formation of inclusions that were reminiscent of what was observed in patients, with the exception of PiD [267]. Indeed, the injections of CBD brain extract caused the formation of silver-positive inclusions similar to the astrocytic plaques noted in CBD, while the aggregates from PSP injections were similar in morphology to tufted astrocytes (hallmark lesions of PSP) [246,267]. Similar findings were observed when the homogenate from the tauopathy cases was injected into non-transgenic mice [268], although there have been differences in the degree and type of the transmitted strain phenotypes noted, which suggest that endogenous host tau isoform expression and perhaps localization influence the strain fidelity [266,271]. When the brain homogenate from injected mice that developed pathologies was re-introduced to naïve mice, again similar pathologies were observed, supporting the presence of tau strains [246]. Furthermore, injections of tau seeds from AD into a neuronal tau knockdown mouse model resulted in little to no neuronal tau propagation and no glial transmission compared to seeds from other tauopathies, which seeded in glial cells, hence confirming further strain-specific cell types [252].

To fully characterize tau strains and provide further validation for the seeding and strain proprieties of tau, various groups developed tau reporter cell lines that express the core-repeat region of tau (tau RD) with two familial mutations fused to YFP [269,272]. The first model, which was developed by Sanders et al. [269], detected the seeding activity of tau, in which tau seeds that transduced to the cells were able to seed the reporter protein and form YFP positive inclusions indicative of tau seeds. Using this model, they also defined two distinct tau strains. One of the tau strains was efficient in seeding the reporter into small internuclear aggregates, while the other led to the formation of large juxtanuclear inclusions [269]. When these two cell-based tau strains were introduced into animals, they induced unique pathologies that were stable even after successive inoculation and formed inclusions when re-introduced to naïve cells that resembled the original inclusions [269].

Afterwards, similar in vitro models were used to isolate and characterized tau strains from various diseases [272,273]. In vitro models have provided an important tool for understanding tau strains as they relate to disparate tauopathies, as shown by the study by Kaufman et al. (2016), in which the model was used for the analysis of 18 tau strains isolated from patients. Their work revealed that strains can be distinguished based on their biological and biochemical proprieties, the brain regions they affect, and the rate of the propagation of pathology, suggesting that tau strains alone may account for the diversity of human tauopathies [273]. Additional biochemical analysis that was conducted by Tanaguchi-Watanabe et al. also supported the same conclusions [169]. One caveat of the aforementioned models is that, in their tau RD-YFP constructs, the bulky YFP is in close physical proximity to the sequences in tau-RD involved in the fibril core formation, which sterically inhibits its templated aggregation into fibrillar tau [274]. Thus, even though multiple tau strains have been described in these models, some caution should be used when interpreting the fibril structures generated in these cells and the nature of the associated aggregation process. Cell models utilizing GFP fusions of full-length tau may better recapitulate the seeded aggregation observed in disease [275]. Despite potential issues with cell models of tau seeding, biochemical analysis of patient tau fibrils by Tanaguchi-Watanabe et al. also supported the existence of distinct strains for individual tauopathies [169]. Moreover, recent structural analysis of tau filaments using Cryo-EM further supports the existence of tau strains by comparing isolates from AD, PiD, CBD, and CTE, wherein tau seemed to adapt distinct conformations in each of these diseases [47,49,57,58]. Further characterization of tau strains is warranted, particularly in vivo. Various aspects of tau strain phenomena remain unknown, such as whether more than one strain can co-exist, what factors may influence the dominance of one strain vs. another, and if all strains share similar mechanisms of spreading or if they vary depending on the disease.

One of the exciting areas is the relationship between PTMs and strains. Can certain PTMs be involved in the encoding of unique strains or affect their properties? If yes, would manipulation of PTMs affect the properties of the encoded strain? The answer to all these questions is still unknown. One interesting hypothesis to examine is that distinct oligomers and fibril conformations are favored by PTM modified tau. In fact, a recent study that mapped PTMs directly onto atomic models of tau filaments from AD and CBD using Cryo-EM and Mass Spectrometry showed that PTMs, specifically ubiquitination could influence the structure of tau filaments contributing to diversity in structure [276]. Further studies about the role of PTMs in mediating tau strain formation and spreading could be enlightening. If tau PTMs are found to promote seeded aggregation or the maintenance of specific tau strains, it raises obvious questions about how those PTMs occur in newly seeded cells. Such mechanisms would imply that the transmitted tau seeds would need to somehow encode for the induction of appropriate PTM modification of unaggregated tau monomers within the new host cell.

Understanding more about tau strains may help in the improvement or development of diagnostic tools for early detection of toxic strains in living patients, which may lead to more accurate or earlier diagnoses. The competition between strains is an interesting area that was recently investigated as one of the new therapeutic strategies for prion disease [277]. One study showed that the introduction of non-toxic or less toxic prion strain protected against the propagation of the toxic strain [277]. This therapeutic approach may be a highly promising application for the prion-like propagation of tau in tauopathies; however, it requires further examination in appropriate models.

### 3.4. Mechanisms and Factors Underlying Prion-Like Tau Transmissions Remain Mysterious

The exact mechanisms of how tau aggregates spread between neurons are still not fully elucidated, nor is the role for other cells (e.g., oligodendrocytes, astrocytes, and microglia) involved in the spreading nor what factors influence or impede prion-like propagation. Various mechanisms of transcellular spreading of tau have been proposed (Figure 3 and Figure 4). The cell-to-cell transmission of tau can occur either transcellularly or by secretion into the extracellular space followed by cellular uptake [278,279] (Figure 3). However, most of the suggested mechanisms that will be mentioned below on tau transmission are supported largely by cell culture studies and remain to be examined/*validated* in vivo. Studying the mechanisms of the prion-like spreading in vivo is essential for better understanding of this phenomena and has multiple advantages. In vivo models can help to validate some of the suggested in vitro mechanisms, unravel new mechanisms, and address questions regarding the prion-like spreading of tau in the complex environment of the living brain (Figure 4). For example, can tau seeds spread via cerebral spinal fluid (CSF), blood, or immune cells such as microglia? How do factors like seizure, sleep/wake, neuronal activity, the glymphatic system, meningeal lymphatic system, and inflammation, each of which can only be faithfully examined in vivo, influence the prion-like spreading?

Appreciating these vectors of tauopathy transmission throughout the CNS can be a rich source of inspiration for innovations in the design of diagnostics and therapeutics.

### 3.5. How Does Tau Spread Intercellularly?

Tau can spread intercellularly to immediately adjacent cells via tunneling nanotubes (TNT) (Figure 3) [278]. TNT are thin filamentous actin-containing extensions of the plasma membrane that connect remote cells, have a diameter of 50–200 nm, and allow the intercellular transport of various cargo including proteins such as prion proteins from neuron to neuron [280,281]. A study by Tardivel et al. (2016) showed that TNT may partly account for tau spreading, since their data showed that both soluble and fibrillar tau transferred from one neuronal cell to another through TNT [278]. However, more information regarding the role of TNT in spreading is required, such as the status of the tau fibrils transferred and whether they are free-form or in vesicles. Ideally, an intervention should be applied that would assess tau propagation when TNT are specifically disrupted. Additionally, in vivo validation is needed, but, considering how thin TNT are, it remains challenging to convincingly study TNT in any sufficiently complex tissue system.

### 3.6. How Does Tau Get Released into the Extracellular Space during Prion-Like Spreading?

Tau can be released outside of cells under physiological conditions without neuropathy and cell death [287,288]. Work by Pooler et al. showed that physiological tau can be secreted into the media during neuronal stimulation [289]. Monomeric tau was found in the brain interstitial fluid (ISF) in wild-type mice, supporting the release of tau under physiological conditions [290]. Additionally, while tau was found in normal cerebral spinal fluid (CSF), there was hyper-phosphorylated tau in the CSF of AD and TBI patients, where its level was higher than normal [291,292]. In addition to ISF and CSF, secreted tau could also be found in blood, although the exact mechanisms of protein transfer to blood are not well understood [293]. However, this transfer may occur through either bidirectional efflux across the brain-blood barrier (BBB) [293]. Studies suggest that tau may transfer from ISF and CSF to the blood via the glymphatic system [204,205] or the dural lymphatic system in the blood [197].

Altogether, it is reasonable to assume that the presence of tau aggregates in extracellular spaces is one of the initial steps involved in the propagation of tau, especially if the tau secreted in pathological conditions is capable of seeding. While the exact mechanism of pathogenic tau release is unknown, recent findings indicate that PTMs such as acetylation may increase the secretion of tau and subsequently contribute to tau propagation [294]. Data from in vitro studies support the proposition that tau could be released extracellularly either in free form or via vesicular-mediated secretory pathways [283].

Tau has been suggested to be released as free soluble forms, through either unconventional mechanisms including direct plasma membrane crossing dislocation and release by secretory lysosomes, or conventional mechanisms, in which the release occurs by the incorporation of protein content in secretory vesicles that pass through the endoplasmic reticulum (ER) and Golgi apparatus then is released via SNARE-mediated exocytosis [283,295]. Evidence of SNARE-mediated exocytosis (the conventional mechanism) was demonstrated first and highlighted that the release of tau depends on the dynamics of Heat Shock cognate 70 (Hsc70) and DnaJ co-chaperone complex as overexpression of DnaJC5, a DnaJ well-known for its ability to stimulate exocytosis, induced the secretion of wild-type and mutant tau [296]. Additionally, recent research has shown that 18 to 28 residues in the N-terminal of human tau serve as a binding motif for End binding proteins (EB proteins), which are implicated in the secretion of proteins such as interleukin 1β (IL1β) [110]. This finding suggests a mechanism by which EB proteins may regulate tau secretion as well as contribute to the propagation of tau pathology.

The possibility of pathogenic tau being released via unconventional cellular mechanisms was raised later after considering the fact that blocking the ER/Golgi-mediated secretory pathway did not block the physiological secretion of tau [287,288]. Taking into consideration the role of the previously mentioned mechanisms of tau release in pathological conditions, it is tempting to think that pathogenic tau could employ the same mechanisms of physiological release of monomeric tau to spread pathology. Indeed, a recent study by Merezhko et al. has shown that phosphorylated oligomeric forms of tau can be secreted via plasma membrane translocation, and this secretion is mediated by heparin sulfate proteoglycans [297]. Interestingly, the tau species secreted via the unconventional mechanisms transcellularly spread to adjoining cells and seeded aggregation. Additionally, the hyper-phosphorylation of tau was shown to increase its secretion extracellularly via membrane translocation [298]. Tau can be released via endo-lysosomal pathways and such secretion is mediated by Rab7, a GTPase that is involved in the trafficking of endosomes, autophagosomes, and lysosomes [299].

Recent evidence has emphasized the role of extracellular vesicles (EVs) in the transfer of pathogenic proteins such as tau in diseases like AD [300,301]. There are two types of extracellular vesicles secreted by mammalian cells that are associated with tau: (1) exosomes which are small single membrane vesicles that are derived from multivesicular bodies with a diameter of 50–100 nm and (2) ectosomes or microvesicles that are shed from plasma membranes with a diameter of 100–1000 nm [302]. Exosomes are an important element of cell communication as they transfer molecules trans-synaptically and intercellularly [303,304]. P-tau was revealed to associate with exosomes in vitro in experiments that overexpress tau [305,306]. Moreover, the analysis of exosomes isolated from CSF of Alzheimer’s patients contained tau protein [305]. A recent study by Wang et al. provided evidence that tau, whether phosphorylated or dephosphorylated, and in monomeric or oligomeric form, can be released extracellularly via exosomes by cultured neurons, N2a cells and in human brains in vivo [307]. The role of microglia and exosomes in the transmission of tau was demonstrated in vivo, in which tau is phagocytosed by microglia and then released via exosomes [251]. Pharmacological inhibition of exosome synthesis in microglia reduced tau propagation. These studies have supported the potential role of exosomes in transmission of tau. However, whether all tau species or strains spread through exosomes is still unknown.

Unlike exosomes, there are a limited number of studies regarding the role of ectosomes in the propagation of tau. Dujardin et al. showed tau can be released in ectosomes under physiological conditions and in the absence of cell damage [308]. Based on the in vitro and in vivo experiments in this study, tau seems to be predominantly secreted via ectosomes under normal conditions; however, during disease conditions, the accumulation of pathogenic tau may cause a shift towards secretion via exosomes or lysosomal pathways [308]. Whether pathogenic tau can be secreted via ectosomes in human CSF, and whether they are involved in the transmission of tau pathology with exosomes are still unanswered questions.

### 3.7. Mechanisms Mediating the Cellular Uptake of Tau Aggregates

With regard to the cellular uptake of tau, various mechanisms have been proposed and supported by a few, mostly in vitro studies [283]. In general, once tau is released it is assumed to enter the cell through the following: (a) fluid-phase endocytosis, which is a bulk uptake of solutes and is a type of macropinocytosis; (b) adsorptive endocytosis that occurs when molecules are binding and concentrating on the cell surface before being internalized; (c) receptor-mediated endocytosis (such as clathrin-mediated endocytosis); and/or (d) uptake of tau when it is in vesicles [283,309]. Tau aggregates were initially suggested to internalize into the cells via fluid-phase endocytosis (macropinocytosis), an active process of the invagination of the plasma membrane that leads to internalization of extracellular fluid, as tau aggregates rather than monomers were taken up and localized with dextran, a marker of fluid endocytosis [218,244,310]. Tau monomers and/or fibrils may be internalized through the involvement of receptors such as APP, M1/M3, or LRP1 [311,312,313]. LRP1, in particular, appeared to play an important role in the cellular uptake of multiple forms of tau, including different tau isoforms and monomeric, oligomeric, and fibril species. In addition, work by Holmes et al. indicated that tau fibrils trigger their internalization by binding to heparan sulfate proteoglycans (HSPGs) on the cell surface, a mechanism also described for prion protein uptake [314,315]. The blocking of HSPGs by heparin both in vitro using human embryonic kidney cells (HEK) and in vivo reduced tau internalization [314]. The 6-*O* sulfate modification on HSPGs is critical for tau fibril uptake, and the enzymatic desulfation of endogenous HSPGs reduced uptake [316].

Recent work by Evans et al. (2018), employing neurons derived from human iPSC, showed that both monomeric and aggregated tau can be internalized into neurons but each used different modes of endocytosis. Monomeric tau can be internalized efficiently by bulk endocytosis while tau aggregates utilized dynamin-dependent endocytosis, a GTPase involved in various fission events, including clathrin-mediated endocytosis and synaptic vesicle endocytosis [317,318]. The role of dynamin-dependent endocytosis was supported by the fact that inhibiting dynamin via the inhibitor dynasore significantly reduced tau aggregate entry by 95% compared to monomeric tau [317]. While this finding is in agreement with another study examining tau aggregation to cultured neurons [319], it was opposite to the findings of Holmes et al., mentioned earlier in this section, in which dynasore did not affect the uptake of tau fibrils [314].

There are various technical issues that may account for these contradictory results that include the cell type and tau species used in each study. In the Holmes et al. study, human embryonic kidney cells (HEK) were used, which are non-neuronal cells and may function differently from human neuronal cells [320]. In addition, Holmes et al. also used tau RD fibrils, composed of amino acids 243 to 375 only, while the other two studies used FL-tau aggregated tau species (recombinant tau oligomers or cell-derived aggregates) [317,319]. Regardless of these inconsistent findings, there is little doubt that dynamin plays an important role in tau pathology; for example, the loss of Bin1, a negative regulator of dynamin, showed enhanced tau pathology [319].

Regarding the uptake of vesicles, in normal conditions exosomes are shown to surf above filopodia before entering cells and being sorted into endosomal trafficking [321]. Whether exosomes carrying tau aggregates use the same uptake mechanism or not remain unknown. The phosphorylation of monomeric tau enhanced its uptake into cortical neurons [322] once again demonstrating the potential of PTMs to influence cellular processes involving tau. Further work employing fibrils or aggregates and their monomer species counterparts isolated from human samples may provide valuable insight towards mechanisms of pathological tau cellular uptake.

### 3.8. Factors Contributing to the Propagation of Tau

As mentioned previously, early in vivo evidence of tau propagation, in which tau propagation affected areas that are anatomically connected, has supported the role of synaptic connectivity and activity in the spreading of tau seeds [323,324,325]. In AD, synaptic connectivity appears to determine the pattern of spreading more than proximity [321,326]. The importance of synaptic connectivity was supported by the recent work of Wang et al. [307]. The study, which utilized a microfluidic device to prevent all mechanisms of tau transfer except synaptic transmission, showed that the depolarization of neurons stimulated the release of tau-containing exosomes, and release was dependent on synaptic connectivity [307]. Additionally, the presence of misfolded tau in both pre- and post-synaptic terminals in AD patients is consistent with the suggestions that tau can transfer over synapses and exerts a pathological role at synapses that may lead to neurodegeneration [327,328,329]. Indeed, pathogenic tau was shown to be able to bind to synaptic vesicles and interfere with their functions leading to synaptic dysfunction [279]. Nonetheless, the precise role of synapses in tau transmission has not been fully explored, particularly in vivo. Additionally, whether tau can utilize synaptic vesicles to be released and spread to other neurons—e.g., during synapse depolarization—is still unknown. However, the trans-synaptic spread of tau via synapses seems to occur before disease-associated synaptic loss and axonal terminal degeneration in tau mutant (human P301L) transgenic mice [330].

The role of the sleep/wake cycle in tau accumulation and its potential in modulating the spreading of pathology are additional areas to consider when studying the prion-like propagation of tau. Sleep is vital for cognitive functions, and the disruption of sleep functions increases the risk of neurodegeneration [331]. Sleep deprivation (SD) is commonly observed among AD patients [331]. In mice, the tau levels in the ISF double after normal waking periods, and this increase is augmented by SD [198]. Intriguingly, sleep-deprived mice also showed increased tau spreading pathology after tau fibril injection. One possible explanation of this finding is that chronic SD can cause the dysfunction of processes that regulate tau ISF release and clearance, including the glymphatic system, which acts as a clearance mechanism primarily during sleep [332]. Tau pathology could synergistically interplay with sleep/wake disruption in a fashion similar to recent proposals for amyloid β oligomers in AD [333]; a feed-forward loop appears likely, where SD increases tauopathy and tauopathy increases SD. Future work should investigate the mechanisms by which sleep disruption can contribute to the prion-like propagation of tau pathology in tauopathies. Appropriate animal models are key for such investigations, because modelling sleep/wake physiology and the fluid dynamics of the glymphatic vasculature is unattainable using in vitro platforms. Intriguingly, SD is also a common symptom of patients following TBI [334,335,336].

In the next section, we consider the tauopathy resulting from TBI, including the prion-like properties that may contribute to CTE. Uncovering factors that modulate the prion-like spreading of tauopathy are expected to inspire the development of new therapeutic approaches and diagnostic tools.

## 4. Prion-Like Properties of Tauopathy Following TBI

In the previous sections, we reviewed the biochemical aspect of tau aggregation and then the prion-like spreading of tauopathies as a key element in the progression of devastating dementias. As a case study in the challenges of attributing prion-like concepts to additional tauopathies, we will briefly review TBI and CTE. We note the potential for prion-like mechanisms to contribute to our understanding of the etiology of these diseases. Complexity arises in part due to the varying types of brain damage caused by different traumatic insults. This complexity may obscure patterns compared to other tauopathies, but may also be leveraged to allow flexibility in the experimental induction of disease.

### 4.1. Overview of TBI

TBIs are defined as a loss of normal brain function caused by insults from external physical forces. An estimated 69 million individuals worldwide sustain a TBI each year, with potentially severe consequences such as disability or reduced life expectancy [337]. TBI is mostly caused by traumatic events such as blows to the head from falls, vehicular collisions, sport-related injuries, or domestic abuse [338,339,340]. It can also result from explosive blasts [341], such as those experienced by military personnel. The severity of the injury at diagnosis is determined by clinical observations such as the occurrence of seizure, hemorrhages, and history such as the duration of loss of consciousness and post-traumatic amnesia [342,343]. Based on these symptoms and the imaging of pathology, TBIs can be classified as mild, moderate, or severe [343]. Mild TBI (mTBI) is a term sometimes associated with concussions, and are among the most common type of TBI observed [344,345]. A TBI can also be categorized as an acute brain injury vs. chronic brain injury. Acute brain injury comprises mTBIs or concussions, including the short-term sequelae of these injuries, and catastrophic brain injuries that may result in death [344]. Chronic brain injury refers to late-term effects, including neurodegeneration and the development of chronic syndromes and diseases [346].

TBI has been linked to increased risk of developing dementias and various neurodegenerative disorders including AD, Parkinson’s Disease, and chronic traumatic encephalopathy (CTE) tauopathies [347,348,349,350,351]. The risk of developing any of these diseases depends on multiple factors, such as the severity of the injury, type of injury (e.g., concussive injury, skull fracture), patient age during the injury (e.g., increased risk with age), and genotype (e.g., APOE4 gene) [83,84,348,352,353,354]. CTE was initially described as “Punch-drunk” and “Dementia Pugilistica” due to its historical association with professional boxers [355,356]. CTE is now more generally linked with repetitive mTBI events that are prevalent in athletes and military personal [354,357].

A significant clinical characteristic of CTE is progressive neurodegeneration, which in advanced cases is associated with cognitive deficits including diminished memory, attention issues, forgetfulness, and dementia [358,359,360]. Impairments in mood and behavior are noted in early periods displayed as mood swings, irritability, and depression, while motor disturbance can develop at later stages of the disease [359,360,361,362,363]. Evidence supports a link between TBI and the development of neurodegenerative diseases (primarily AD and CTE). However, skepticism is still warranted because most of the data supporting this link are from retrospective studies that are susceptible to incomplete or biased methods and clinical data [364]. Moreover, the mechanisms by which head trauma induces neurodegeneration remain poorly understood. Thus, there is pressing need in the field for prospective longitudinal studies of TBI patients. These could provide reliable evidence of the neuropathological risks associated with head trauma and identify clinical and biological markers to track the associated disease progression [365]. These prospective studies should consider the type, location, and severity of the brain injury; include post-injury treatments/patient care; and examine the short and prolonged effects of TBI on cognitive functions through various approaches such as neurological tests and patient bio-samples. They should also evaluate the impact on brain structures over time via brain imaging technologies, some of which could also evaluate changes in proteins involved in neurodegeneration, such as the PET imaging of tau pathology [366]. It is also important that experimental models of TBI be developed in parallel to clinical studies to lend insight into the molecular mechanisms underlying TBI and the validated neuropathology risk.

With respect to the biological effects of TBIs, they are known to directly result in damage to nervous tissues and perturb normal brain function [343]. The primary injury causes tissue damage directly through physical force. It can occur via the rapid acceleration and de-acceleration of linear forces on the head, direct blunt impact, or by forces (shockwaves) generated by the blast wind associated with blast injury [343,367]. The primary insult then elicits multiple biochemical, cellular, and molecular alterations that perturb the CNS environment and cause secondary brain injuries that can develop days, weeks, or years after the TBI event [345]. Major alterations described include the loss of ionic homeostasis combined with the release of excitatory neurotransmitters, vasculature abnormalities and disruption of the blood–brain barrier (BBB), neuroinflammation, and alterations affecting the cytoskeleton [368,369]. The involvement of tau pathology has previously been included among the many hypotheses behind this ongoing injury in TBI [370,371,372]

In 1973, tau aggregates and filaments were first observed and reported in the post-mortem analysis of boxers’ brains [373]. Since then, the pathological association of tau deposits in CTE and after repetitive mTBIs has become the most prominent neuropathological feature found in post-mortem in CTE and TBI patients [264,344,374,375]. In CTE, tau pathology can be observed as NFTs, as clusters in the depth of cortical sulci, and as inclusions in astrocytes around blood vessels [264]. Linking TBI to potential tau alteration, the serum levels of tau were elevated at 2 and 7 days post-TBI [376,377]. These studies were supported by an in vivo rodent study showing transient elevated serum levels of tau that were measured as early as one hour post-TBI and positively correlated with the severity of the brain trauma [378]. Other studies in animal models of TBI and repetitive mTBIs also reported similar short-term changes in tau, with noticeably increased tau immunoreactivity and the detection of multiple p-tau species within 24 h [371,372]. In contrast, a histopathological examination of the long-term effects of mTBI on tau alterations revealed extensive NFT formation after 9 months, which was associated with behavioral deficits [379]. It is important to note that tau changes are not detected in all repetitive TBI animal models, and more work is needed to resolve the basis of such discrepancies (e.g., injury protocol, genetic background).

CTE is linked to repetitive TBI events, but it is clear that single TBI events can also trigger tau pathology. Long-term survivors of single moderate to severe TBI showed the presence of a wide and abundant distribution of NFTs with a similar distribution of tau aggregates as those associated with repetitive mTBI [380]. A similar recent study by Gorgoraptis et al. [366] has shown the presence of tau pathology in TBI patients years after being subjected to a single brain trauma using tau PET technology. This revealed a correlation between tau pathology and long-term neuronal damage associated with TBI compared to healthy controls [366]. Since single TBIs have been linked to increased dementia risk but not CTE, it is possible that the tau pathology pathways are distinct [337]. Although still relatively unexplored, researchers are starting to directly compare the effects of repetitive TBI vs. single TBI events on tau alterations. They have demonstrated differences in both the induction of p-tau and oligomer characteristics between single and repetitive injury regimes [381,382]. In the next section, we will explore whether alterations in tau protein and its prion-like propagation might cause secondary injury and CTE risk.

### 4.2. Prion-Like Proprieties of Tau in TBI and CTE

Multiple observations now demonstrate: (1) early molecular change in tau with repetitive or single TBIs; and (2) tau aggregation pathology in CTE. Can these disparate events be linked via prion-like spreading mechanisms? Convincing evidence would include TBIs inducing tau aggregation and spreading between cells and CNS tissues, as well as faithful templating of tau strain in disease.

Similar to other tauopathies, tau pathology in CTE seems to be confined in specific areas early in the disease progression. Perivascular glia and neurons containing p-tau appear in the cortical sulci, and the pathology spreads at later stages to cortical, medial temporal, and subcortical grey matter [46,374,383]. The finding of a characteristic route of tau pathology accumulation correlating with disease severity is reminiscent of other tauopathies associated with prion-like spread such as AD [41]. In vivo mouse TBI studies have supported the initiation and spread of tau aggregation and pathology [380,384]. In a study, transgenic mutant tau (P301S) mice were subjected to moderate to severe TBI and showed an increase in tau aggregation and the acceleration of the pathology compared to sham mice post-injury [384]. Further, the pathology spread to synaptically connected regions; the authors postulated that TBI could increase the risk of developing tauopathies such as CTE through the induction of tau aggregation events [384]. Zanier et al. found that a single severe TBI in non-transgenic mice triggered p-tau pathology that spread throughout the brain up to at least 12 months post-injury [380]. Additionally, TBI induced synaptic loss and memory deficits. Although they did not directly assay for tau aggregates, they did show that the tau in TBI mice showed protease resistance, which is characteristic of aggregate formation. Zanier et al. went further to show that the introduction of brain homogenate from TBI mice to naïve mice induced similar tau pathology, synaptic loss, and memory deficits. In vivo work by Gerson et al. showed that tau oligomers isolated from rats subjected to two methods of TBI (blast-induced injury and fluid percussion injury) and injected into mice expressing human tau induced seeding and aggregation in the injection site and other areas [213]. These tau species caused cognitive deficits and enhanced pathology. In addition, experiments in a YFP-tau reporter cell model have demonstrated that tau aggregates isolated from CTE patients successfully seed tau aggregation in these cells [272]. Together, these results support the concept that TBI-induced tau pathology and toxicity can be faithfully propagated in a prion-like manner via tau aggregate species.

Despite mounting evidence supporting the prion-like proprieties of tau seeds formed after TBI, key knowledge gaps still exist. In particular, the factors and mechanisms governing the spreading of tau pathology between different regions in TBI are not yet known. We propose that secondary injury events that occur in the brain in response to TBIs can promote tau pathology spread. As an example, the glymphatic system has been shown to be important for the clearance of extracellular tau [205,385]. A recent study reported a 60% reduction in glymphatic pathway function after TBI [205]. Documented post-TBI sleep disruptions may further compromise glymphatic system function [386,387].

Given the prevalence of post-traumatic seizures (PTS) and post-traumatic epilepsy (PTE) in patients with blast-related TBI [388], the roles of neuronal excitability in the sequelae of TBI (including whether it is involved in tau pathology spreading or not) are also of interest. In a recent study in veterans, 57% of seizures observed were linked to TBI [389]. Early post-traumatic seizures (occurring within the first week after the brain injury) are seen in the majority of cases of TBI (both adults and children), and their occurrence and frequency are associated with the severity of the injury [390,391]. PTS and PTE are both major contributing factors to increased morbidity and mortality risk for several years following the TBI [390,392]. Thus, the prophylactic application of anti-convulsants is used in some TBI cases to prevent acute post-traumatic seizures after TBI [393]. At the molecular level, the TBI disruption of microRNAs promotes the release of excitatory neurotransmitters (e.g., glutamate) that may create cellular conditions that are more susceptible to developing seizures and epilepsy [394,395]. There is some evidence linking TBI, seizures, and tau pathology processes. Tau phosphorylation and aggregation have been observed in patients with late-onset epilepsy several years post-injury [396]. Interestingly, there are multiple reports that seizures alone cause tau biochemical changes; tau phosphorylation is increased in animal models of chemically or electrically induced seizures [397,398,399,400]. However, more information is needed regarding the exact mechanism by which seizure could impact or intensify tau propagation after TBIs. Information regarding the effects of different types of seizures (convulsant vs. absent or non-convulsant) on tauopathy progression and neurodegeneration after post-traumatic brain injury is of great interest and may be of therapeutic value. A critical experiment would be to test whether anti-convulsants ameliorate the tau pathology in animal models of TBI that produce seizure phenotypes [401]. It would also be interesting to compare the tau prion-like spreading and pathology in naïve mice inoculated with brain homogenates from various seizure models (TBI-induced, chemically induced, genetic absent seizure) [401,402,403]. Anti-convulsant treatment groups would also augment these propagation studies. These experiments could provide a clear answer as to whether altered neuron excitability is a major driver of tau prion-like pathology post-TBI and whether long-term studies of preventative anti-convulsant treatments are warranted. The TBI and larger tauopathy community may also benefit from additional experimental models that can track tau aggregate propagation through tissues in real time. Several radio imaging tracers are currently being developed to follow tau pathology over time, in the clinic (reviewed in [404]). It would be very powerful to optimize parallel tracking experiments using the same or similar tracers in rodent models of tauopathies such as TBI [405]. We can also take advantage of other animal model systems to study the prion-like aspects of TBI. For example, we have recently generated a fluorescent tau reporter model in zebrafish that allows us to directly observe tau aggregate formation and its subsequent movement throughout the nervous system following a TBI [406]. This model has the potential to elucidate if and how tau propagation spreads from one brain region to another and determine how variables such as injury severity, sleep disruption, or neural activity modulate these spreading patterns.

### 4.3. CTE May Be a Unique Prion-Like Strain of Tauopathy

Although not yet conclusive, multiple lines of evidence support the notion that CTE may represent at least one unique tau strain. CTE appears to generate distinct tau properties compared to other tauopathies, as suggested by post-mortem observations of: regions and cells affected by tau pathology; seeding properties and spreading patterns; isoform composition of tau in filaments; and structural diversity of tau filaments [57,264,272,407,408,409]. With respect to the tau pathology in CTE compared to that in AD, for instance, the perivascular distribution of NFTs in CTE is found to affect both neurons and astrocytes concentrated preferentially at the superficial layers of the cortical sulci (layer II and III), which is strikingly different from the NFTs in AD which affect neurons predominant at the deep cortical layers, with a higher density in layer V and VI [264,408]. Furthermore, the tauopathy progression in CTE appears as an irregular widespread pathology that varies from the diffuse spread of tau pathology in AD brains [264,409]. Interestingly, recent work by Woerman et al. has shown that the seeding potential of the patient-derived tau aggregates in AD and CTE could be distinguished from that of other tauopathies (PiD, AGD, CBD, and PSP) using a panel of tau-YFP reporter cell models [272]. Although the exact molecular mechanisms driving the detection differences between cell lines still need to be worked out, the data support that CTE tau aggregates have some feature that drives a distinct mechanism of pathology. However, this study did not provide a detailed analysis of the morphology or localization of the inclusions formed in the reporter-expressing cells that have been employed in similar cell models [269,273]. Such an analysis might reveal differences between the CTE and AD seeding profiles to further support CTE aggregates as a distinct tau strain. In this regard, recent data obtained from examining the tau filaments in CTE supported the structural diversity of tau fibrils in this disease compared to AD and other tauopathies [57]. In this work, the authors examined tau filaments isolated from three CTE patients and found that approximately 90% of the tau filaments, denoted as Type I, coming from these cases are structurally different from the paired helical filaments (PHFs) and straight filaments (SFs) found in AD and filaments in other tauopathies such as Pick’s disease [47,57,58]. Negative stain EM analysis indicated that these CTE-tau filaments are composed of C-shaped proptofilaments made of 3R and 4R domains, reminiscent of the filaments in AD, but they also contain a more open C-shaped conformation, with an additional hydrophobic cavity (density) which is not seen in AD. Unfortunately, the authors were unable to propose a specific cellular factor to account for this density, given the structural data on hand. The remaining 10% of CTE-tau filaments, called Type II, resembled PHFs but contained the additional density seen in Type I. Interestingly, even though both Type I and Type II shared indistinguishable protofilaments with an additional density, the interfaces between the protofilaments were different, indicative of small changes (polymorphs) at the ultrastructural level [57]. This data is a strong validation that the tau filaments in CTE can be classified as independent and potential multiple distinct tau strains. It would be interesting to see if the distinct CTE strains could be recapitulated in vitro and could be compared for their propagation properties both in biochemical assays and animal models.

A related outstanding question is how many strains may be formed post-TBI and whether there is strain competition that can be pushed in one direction or another to drive CTE or other long-term outcomes. Is it possible that the specific parameters of individual TBIs discussed earlier in this review (the physical nature of the injury, sleep disruption, treatments) have any influence on the number and characteristics of the tau strains that develop? Which of these tau strains could we attribute to driving spread and neurodegeneration, and would they ultimately look like the aggregates found in post-mortem CTE patients or would they be distinct, suggesting that only early treatment interventions would be effective in mitigating CTE risk? To examine these questions, future experiments should be designed to implement the manipulation of TBI characteristics, including intensity (mild vs. severe), methods of brain injury (blast vs. weight drop), and number of traumatic injury (single vs. multiple), followed by an in-depth analysis of the tau species formed after each condition. The analysis should include looking at the structural differences in the tau filaments formed, assessing the efficiency of the tauopathy seeding (via the use of in vitro models such as [269,273]), examining the rate of tauopathy spread, and evaluating the biochemical properties of each tau species. One of the commonly assessed biochemical properties employed to distinguish between distinct strains for prions as well as tau is examining the size of protease-resistant core fragments [273,410,411]. When soluble tau converts into insoluble aggregates, it acquires resistance to proteases. Thus, the sizes of protease-resistant core fragments are used as “fingerprints” representing distinct tau strains [273,410]. The proposed experiments and analyses would permit a better understanding of the landscape and associated pathology of the tau stains generated by TBI.

## 5. Conclusions

Considerable work has been directed into understanding the physiological and pathological functions of tau protein in tauopathies. The observations and data collected from diverse sources, including patients, cells, and animal models, have all led to the generation and support of the prion-like hypothesis of tauopathy progression. The number of studies supporting the prion-like propagation of tau, especially in AD, keeps growing, and our work highlights the need for parallel work in other tauopathies, such as TBI and CTE. Considerable work is still needed to understand the biological drivers and consequences of the heterogeneity of tau aggregates observed among those diseases. Additionally, we need to continue to develop experimental models of TBI to reveal information regarding the molecular mechanisms underlying the prion-like propagation of tau pathology for each individual tauopathy. Information defining tau strains and their associated potential to promote pathogenesis in the CNS could pave a route to identifying cellular targets for therapeutic interventions.

## Figures and Tables

**Figure 1 biomolecules-10-01487-f001:**
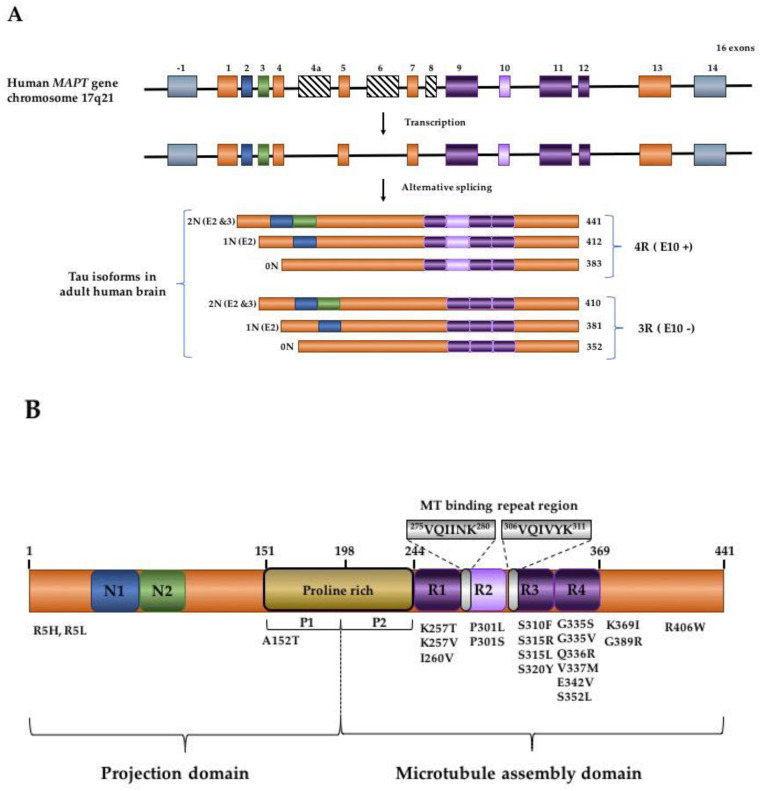
The human *MAPT* gene, tau isoforms in the human brain, structure, and mutations. (**A**)**.** Schematic of the *MAPT* gene on chromosome 17q21.31 which compromises 16 exons [10,31]. There are six main isoforms of tau that are generated from the alternative splicing of E2, E3, and E10. The splicing of E2 and E3 generates isoforms containing either 0, 1, or 2 amino-terminal inserts of 29 amino acids known as 0N, 1N, and 2N, respectively. The presence or absence of the second repeat R2 domain (light purple), which is encoded by exon 10, categorize the isoforms with 3R or 4R [32]. (**B**). Tau major domains are divided into the projection domain and the assembly repeat domain in the carboxy-terminal sections separated by a proline-rich region. The projection domain comprises residues 1–197 and is not directly involved in microtubule (MT) binding. The proline-rich region is subdivided into P1 and P2, separated by the chymotryptic cleavage site at residue 198 that divides the assembly and projection domain [33,34]. The C-terminal assembly domain is important for MT binding and assembly. The assembly domain contains the MT binding repeat region followed by a flanking region that shows a weak sequence similarity to the repeat domain [33]. The four repeats, around 30–31 residues each, are labeled R1-R4. Both P2 and the flanking regions contribute to MT assembly and binding. The MT repeat region also contains the “paired-helical filament core”, which serves as a principal structure for forming tau aggregates [35,36]. Within this structure, two hexapeptide motifs [37] important for aggregation have been highlighted in gray. Major disease-associated missense mutations that alter the sequence are also labeled.

**Figure 2 biomolecules-10-01487-f002:**
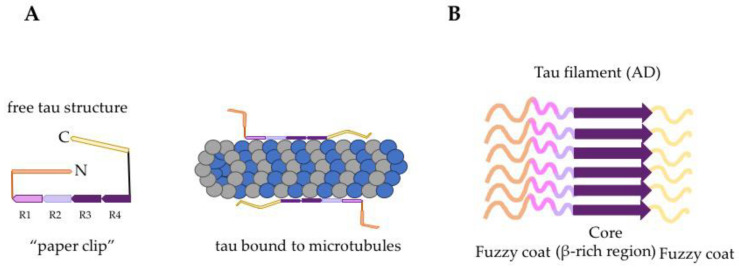
Possible tau structures. (**A**) The proposed “Paper-clip” structure of tau, in which both the N and C terminals are closely associated [141]. However, when tau is bound to microtubules, the two terminals are separated with the *N*-terminal projected away from the microtubules [124]. (**B**) In AD tau filaments, part of the tau repeat region (R3 and R4) forms the core of the filaments while R1, R2, and both N and C terminals form the “fuzzy coat” structure that surrounds the core (based on the Cryo-EM structure described in [58]).

**Figure 3 biomolecules-10-01487-f003:**
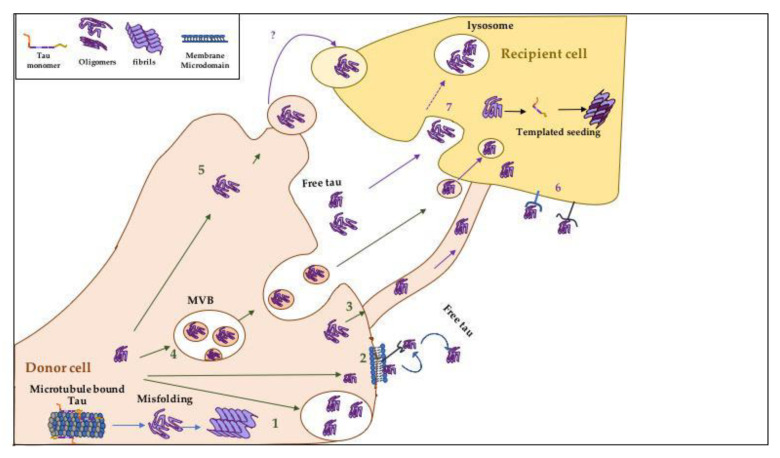
Various hypothesized mechanisms involved in the prion-like spreading of tau pathology between adjacent or nearby cells (inspired by figures in [282,283]). The mechanisms include the spread of tauopathy seeds between adjacent cells. (A) Tau aggregates inside neurons can spread from the donor (pre-synaptic) to recipient neurons (post-synaptic) via various mechanisms. Donor cell: (1) Regulated release via either soluble *N*-ethylmaleimide-sensitive factor attachment protein receptor (SNARE)-dependent exocytosis, endo-lysosomal pathways mediated by Rab7, or EB proteins; (2) direct secretion via plasma-membrane (PM) translocation that includes the clustering of tau at the PM, interactions with specific lipids and release from the PM guided by HSPG cell-surface receptors; (3) intercellular transfer between neurons via tunneling nanotubes; (4) the fusion of multivesicular bodies (MVB) to plasma membranes and secretion of tau aggregates in exosomes; (5) the release of tau aggregates in ectosomes. At the recipient cell(s), released tau aggregates can be taken up by various mechanisms, including (6) receptor-mediated endocytosis (HSPGs and APP), (7) macropinocytosis, or dynamin-dependent endocytosis. The process by which cells take up exosomes and ectosomes is still unclear.

**Figure 4 biomolecules-10-01487-f004:**
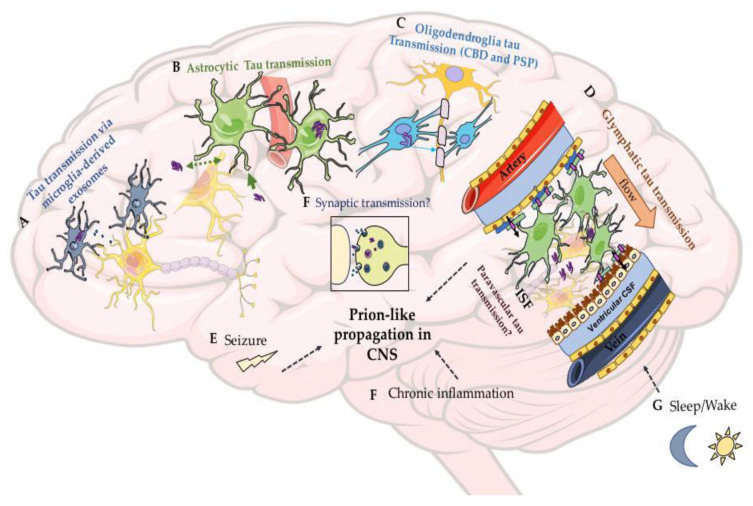
Various hypothesized mechanisms and vectors of the prion-like spreading of tau pathology between tissues. (**A**) Tau can be phagocytosed by microglia, transported, and then released via exosomes, which contribute to the spreading of tau pathology. Tau can be internalized by non-neuronal cells, such as (**B**) astrocytes or (**C**) oligodendrocytes, which contribute to glial tau transmission in tauopathies other than AD such as CBD and PSP. (**D**) One of the recently studied mechanisms that may account for the spread of tau pathology is the glymphatic system’s involvement (the structural organization of panel D is inspired by schematics in [284,285,286]). In this system, free tau can be cleared from the ISF using the CSF influx into the extracellular spaces within the brain parenchyma. The CSF influx flows directionally through the aquaporin four (AQP4) channel (colored in pink) [205] that is highly expressed in the end-feet of astrocytes lining the arterial and venous perivascular spaces (described in [204]). Various factors that may affect tau spreading and aggregation also need to be considered, including (**E**) the role of seizures, (**F**) synaptic connectivity and transmission, and (**G**) sleep/wake cycles and disruptions. Investigating these mechanisms, vectors, and factors impacting tauopathy progression is a priority area and urgently requires improved animal models of disease.

**Table 1 biomolecules-10-01487-t001:** Pathological features of major tauopathies (some information obtained from [39,46]).

Common Tauopathies	The Form of Tau Aggregates	Tau Filaments Composition and Ultrastructure	Supportive Pathological Features Not Related to Phosphorylated Tau (P-Tau)	Cells Affected by Tau Pathology and Disease Progression
**Pick’s Disease (PiD) **(primary tauopathy)	Round cytoplasmic tau aggregates (known as Pick bodies), ramified astrocytic filaments, and neuropil threads.	Filaments consist of 3R tau isoforms. Most of tau filaments (93%) are narrow filaments (with a slight helical twist), previously mentioned as straight filaments (SF), and the remaining small percent are classified as wide filaments [47].	Neuronal loss, gliosis, and ballooned cells known as Pick cells	Tau pathology is observed in both neuronal and glial cells. Tauopathy is found restricted to neocortical and limbic areas in the frontotemporal regions, then spreads to subcortical structure and brainstem, followed by the motor cortex. In severe cases, it is found in the visual cortex [48].
**Corticobasal Degeneration (CBD) **(primary tauopathy)	Filaments take the shape of comma or coil-like (coiled bodies) pre-tangles, astrocytic plaques, and neuritic threads.	Tau filaments composed of 4R isoforms. The filaments in CBD are heterogeneous, containing narrow single-stranded filaments and wide double-stranded filaments, with the first being three times more abundant [49,50].	Neuronal loss and degeneration of the substantia nigra in mild and severe cases.	Tau pathology affecting both neurons and glia. Based on a recent neuropathological analysis of CBD cases, tau pathology is limited to the anterior of the frontal cortex, amygdala, and basal ganglia in earlier stages, then progresses to affect more prominently the frontal and parietal cortices, amygdala, caudate, subthalamic nucleus, and pontine tegmentum in late stages [51].
**Progressive Supranuclear Palsy (PSP) **(primary tauopathy)	Neurofibrillary tangles, globose tangles, coiled bodies, and star-like or tufted astrocytes.	Filaments consists of 4R tau isoforms. Tau filaments formed are SF with rare twisted filaments [52].	Neuronal loss and gliosis.	Tauopathy affects both neurons and glia. Pathology is restricted in an earlier stage to the pallido-luyso-nigral regions then proceeds to involve the basal ganglia. In later stages, the pathology progresses to involve areas in the frontal and parietal lobes such as the subthalamic nucleus and the substantia nigra [53].
**Globular Glial Tauopathy (GGT) **(primary tauopathy)	Large and globular tau inclusions in oligodendrocyte and astrocytes.	Filaments composed mainly of 4R isoforms. Information regarding the ultrastructure of tau filaments is still unknown.	Atrophy, neuronal loss, and gliosis in the primary cortex and or in the corticospinal tract (some cases).	Tau filaments found in both neurons and glial cells. Rare 4R tauopathy and cases present with different patterns of neuropathology depending on the affected area. In a subtype of GGT that causes motor neuron disease, tau pathology is observed throughout the primary motor cortex. In a second subtype, which is associated with frontotemporal dementia, tau pathology is detected throughout the frontotemporal cortex [54]. Further information regarding the progression of tauopathy is still needed in GGT.
**Argyrophilic Grain Disease (AGD) **(primary tauopathy)	Argyrophilic grains, oligodendritic coiled bodies, neuronal pretangles.	Tau filaments composed of 4R isoforms.	Spongy ambient gyrus degeneration.	Tau filaments affects both neurons and glia. Tauopathy affects the ambient gyrus initially and then progresses through the medial temporal lobe. Tau pathology in advanced cases seen in septum and insular cortex [55].
**Chronic Traumatic Encephalopathy (CTE) **(primary tauopathy)	Neurofibrillary tangles, dot-like or grain like neurites, and astrocytic tangles and thorn- shaped astrocytes	While neuronal tau filaments consist of 3R and 4R isoforms, astroglia tau filaments are composed mainly of 4R isoforms [56]. 90 percent of the filaments are helical filaments called type 1 which are distinct from PHFs and SF found in AD. The remaining are similar to PHFs in AD [57].	TDP-43 Neuronal and glial Inclusions in severe cases. Axonal injury and degeneration Diffuse Aβ plaques (present in ~50% of cases) [46].	Tau aggregates affect both neurons and glia. The pathology begins in the frontal cortex around small vessels then spreads throughout the cortex. The pathology affects the outer layer of the cerebral cortex, unlike AD (more details mentioned in Section 3.4 and 3.5)
**Alzheimer’s Disease (AD)** (secondary tauopathy)	Tau filaments in the form of neurofibrillary tangles, neuropil threads.	Tau filaments composed of both 3R and 4R isoforms and found as paired helical filaments (PHFs) and SFs [58].	Progressive accumulation of Aβ plaques in distinct patterns (one of the main hallmarks of AD) [59].	Tau pathology seen predominantly in neurons. It starts at the entorhinal and transentorhinal regions, then progresses to the limbic regions including the hippocampus and, at later stages, tauopathy is observed throughout the neocortex [41].

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
