# Peer review of "Prion-Like Propagation Mechanisms in Tauopathies and Traumatic Brain Injury: Challenges and Prospects"

_biomolecules, 2020, doi:10.3390/biom10111487_

Round 1

Reviewer 1 Report

The present manuscript is a well written, interesting and very comprehensive  review on tau physiology.

Author Response

We thank you for taking the time to review our manuscript.

Reviewer 2 Report

Despite the great changes that have occurred in this new version of the manuscript (including the order of the authors), I maintain my positive opinion regarding the potential of this review in the field of TBI and CTE.

Personally, I would order some of the new sections differently. In this sense, the functions of the protein (point 2.4) should be explained prior to tauopathies (point 2.2). In addition, I consider that the new expansion of sections 2 and 3 has fallen into the repetition of some aspects, which could have been further summarized. However, the extension of section 4 has considerably improved the revision, being the main object of this text. For all the above, I only suggest to authors check:

Some of the minor points that I suggested in the in the previous revision of this manuscript that remain unresolved:

-Symbols for MAPT gene must be italicized (line 252)

-The use of the term “vs.” is still different throughout the text: vs, vs. versus…See for instance lines 629, 656 and 1047.

-The term “tauopathy” is written interchangeably as “tauopathy” and “Tauopathy” throughout the text (e.g. lines 70, 77, 146…). Please revise.

- The term “tau” is written interchangeably as “tau” and “Tau” throughout the text (e.g. lines 213 and 215). Please revise.

-Line 64: Delete “Traumatic brain injury” as this is not the first mention. Revise along the text.

Additional minor points:

-Line 110: The final parenthesis after the reference (23) is missing.

-Line 135: Ending period is missing after “(see section 2.3)”.

-Lines 150 to 171: Revise font size and style of the figure legend. The same in various paragraphs of the text (e.g. lines 308-328, 1528-1548).

-The terms Fontotemporal lobar degeneration (FTLD) and Frontotemporal dementia (FTD) are used interchangeably (e.g. lines 192 and 269). Please check the text and first mention in each case.

-Lines 284-287: In reference to tau aggregation propensity in animal models, please rewrite the paragraph.

Line 289: Use the symbols “–“ or “/” between genotype and phenotype instead of “:”?

-Line 426: Lack of space between “(Figure 2B)” and references.

-Line 637: Revise the term (extra)cellular.

-Line 657: Change “post-translation” to “post-translational”

-Line 681: The abbreviation for phosphorylated tau; phospho-tau (p-tau) must be indicated in the first mention and use only one abbreviation along the text (phospho-tau or p-tau) (e.g. lines 1645 and 1656).

-Line 801: Write “PrPSc” instead “PrPsc”.

-Line 804: It would be more clarifying if sCJD were introduced as an example of human prionopathies showing different strains. Please rewrite this sentence.

-Lines 1554 y 1555. Rewrite the sentence to avoid repeating “The primary injury”.

-Line 1616: Change eg. to e.g.

Author Response

We thank the reviewer for taking the time to provide a thorough review of our manuscript. Please see below for our revisions to address your comments individually.

Please note that our line count did not match yours in the revised manuscript but we believe we have found all the equivalent lines to change and have used those for reference.

Despite the great changes that have occurred in this new version of the manuscript (including the order of the authors), I maintain my positive opinion regarding the potential of this review in the field of TBI and CTE.

There was a mix up in the author order generated during submission. Thank you for pointing out the error. It has been fixed.

Personally, I would order some of the new sections differently. In this sense, the functions of the protein (point 2.4) should be explained prior to tauopathies (point 2.2). In addition, I consider that the new expansion of sections 2 and 3 has fallen into the repetition of some aspects, which could have been further summarized. However, the extension of section 4 has considerably improved the revision, being the main object of this text. For all the above, I only suggest to authors check:

Some of the minor points that I suggested in the in the previous revision of this manuscript that remain unresolved:

-Symbols for MAPT gene must be italicized (line 252)

All instances fixed throughout section 2.

-The use of the term “vs.” is still different throughout the text: vs, vs. versus…See for instance lines 629, 656 and 1047.

All instances fixed to “vs” throughout the manuscript e.g. lines 179, 476, 851

-The term “tauopathy” is written interchangeably as “tauopathy” and “Tauopathy” throughout the text (e.g. lines 70, 77, 146…). Please revise.

All instances changed to “tauopathy” unless at the beginning of a sentence. Fixed in Table 1 and on lines 120, 177, 430, 466

- The term “tau” is written interchangeably as “tau” and “Tau” throughout the text (e.g. lines 213 and 215). Please revise. 

Fixed currently at line 170, 409, 444 and in table. The term is left written as “ Tau” when it is at the beginning of the sentences.

-Line 64: Delete “Traumatic brain injury” as this is not the first mention. Revise along the text.

 Deleted on line 48

Additional minor points:

-Line 110: The final parenthesis after the reference (23) is missing.

Added on line 92

-Line 135: Ending period is missing after “(see section 2.3)”.

Added on line 111

-Lines 150 to 171: Revise font size and style of the figure legend. The same in various paragraphs of the text (e.g. lines 308-328, 1528-1548).

All figure legends now have the same font size and style (Palantino Linotype, size 9)

-The terms Fontotemporal lobar degeneration (FTLD) and Frontotemporal dementia (FTD) are used interchangeably (e.g. lines 192 and 269). Please check the text and first mention in each case.

Revised text to include FTLD term in line 221

-Lines 284-287: In reference to tau aggregation propensity in animal models, please rewrite the paragraph.

The sentence was re-written starting on line 235. “Multiple MAPT mutations have been shown to increase the propensity of tau protein to seed aggregation compare to WT resulting in the development of tau pathology in animal models [78-82].”

Line 289: Use the symbols “–“ or “/” between genotype and phenotype instead of “:”?

Corrected to “-“ on line 238

-Line 426: Lack of space between “(Figure 2B)” and references.

Fixed on line 351

-Line 637: Revise the term (extra)cellular.

Revised line 508

-Line 657: Change “post-translation” to “post-translational”

Revised on line 509

-Line 681: The abbreviation for phosphorylated tau; phospho-tau (p-tau) must be indicated in the first mention and use only one abbreviation along the text (phospho-tau or p-tau) (e.g. lines 1645 and 1656).

Fixed in Table 1and line 313, the abbreviation p-tau was used throughout the document and the (phospho-tau) was deleted from line 1009 and 1233.

-Line 801: Write “PrPSc” instead “PrPsc”.

Fixed in line 638 and 644

-Line 804: It would be more clarifying if sCJD were introduced as an example of human prionopathies showing different strains. Please rewrite this sentence.

Revised on Lines 640-643. “Evidence of PrPSc strains can be observed for the most common prion disease, sporadic Creutzfeld-Jakob Disease (sCJD). sCJD has a heterogeneous clinical and pathological presentation and individual cases of sCJD can be further categorized into one of six subtypes [229].”

-Lines 1554 y 1555. Rewrite the sentence to avoid repeating “The primary injury”.

Revised on line 1197

-Line 1616: Change eg. to e.g.

Fixed in line 1225

Reviewer 3 Report

This is a very extensive and comprehensive review of prion-like propagation mechanisms in tauopathies and neurodegeneration caused by traumatic brain injury (v.g. dementia pugilistica). It also includes an extensive background on the genetics and biochemistry of tau and includes a very complete list of references. It will be of interest to researchers in the field of protein misfolding, tauopathies and neurodegenerative diseases. A few minor changes would in my opinion improve the manuscript:

  • In the title, TBI should be spelled out: Traumatic brain injury (TBI)…
  • In line 567, it is stated that PrPC is ubiquitously expressed. This is not correct: its expression is limited to the brain and to a much lesser degree, a few other organs.
  • In the paragraph about biology and propagation of PrPSc and prion diseases, there is substantial ambiguity about the “strain” concept. In prion biology, “strain” is only applied to different subtypes of PrPSc, not different subtypes of prion disease. As an example, sporadic Fatal Familial Insomnia and sporadic Creutzfeldt-Jakob disease are two distinct prion diseases, with different clinical phenotypes, caused by different PrPSc strains. But they are not two strains of prion disease.
  • In the same paragraph, a PrPsc with sc as subscript should be changed to Sc as superscript.
  • Throughout the manuscript, the concept “prion-like” is extensively used. The term has a very substantial currency, but there also is a controversy in the field with three points of view: some authors contend that the term aptly describes the characteristics of tau, Abeta, a-synuclein…propagative aggregates; others, prefer the term “prionoid”, and finally, a third group supports the notion that these are in fact full fledged prions. The main point of contention is the relative infectivity/transmissibility of the various propagative aggregates: several PrPSc strains are clearly infectious and have caused epizootics and epidemics, like mad cow disease, scrapie or Kuru, while others, tau aggregates for example, can be transmitted between individuals under very extreme conditions. The Authors might want to cite Eraña H., Prion. 2019 Jan;13(1):41-45. For a very complete and up to date discussion of this controversy.

Author Response

We thank the reviewer for taking the time to provide a thorough review of our manuscript. Please see below for our revisions to address your comments individually.

Please note that our line count did not match yours in the revised manuscript but we believe we have found all the equivalent lines to change and have used those for reference.

This is a very extensive and comprehensive review of prion-like propagation mechanisms in tauopathies and neurodegeneration caused by traumatic brain injury (v.g. dementia pugilistica). It also includes an extensive background on the genetics and biochemistry of tau and includes a very complete list of references. It will be of interest to researchers in the field of protein misfolding, tauopathies and neurodegenerative diseases. A few minor changes would in my opinion improve the manuscript:

In the title, TBI should be spelled out: Traumatic brain injury (TBI)…

The abbreviation was spelled out as requested

In line 567, it is stated that PrPC is ubiquitously expressed. This is not correct: its expression is limited to the brain and to a much lesser degree, a few other organs.

Correct on line 599-600. “The cellular form of prion protein (PrPC) is expressed mainly in the brain and is characterized as a ubiquitous glycoprotein largely localized to the plasma membrane in cells.”

In the paragraph about biology and propagation of PrPSc and prion diseases, there is substantial ambiguity about the “strain” concept. In prion biology, “strain” is only applied to different subtypes of PrPSc, not different subtypes of prion disease. As an example, sporadic Fatal Familial Insomnia and sporadic Creutzfeldt-Jakob disease are two distinct prion diseases, with different clinical phenotypes, caused by different PrPSc strains. But they are not two strains of prion disease.

Text revised on lines 633-640.  “Another defining characteristic of prion pathobiology is the concept of “strains”. The strain concept was developed based on data that PrPSc isolated from a given source (strain) and inoculated into naive hosts consistently resulted in a characteristic profile of clinical phenotypes, pathology and PrPSc biochemical properties. The differences in the biochemical properties of individual strains such as resistance to protease digestion or denaturing agents [228] has led to the current accepted view that strains can be separated based on structural differences in their PrPsc aggregates that dictate seeding and pathogenic effects in vivo. “

In the same paragraph, a PrPsc with sc as subscript should be changed to Sc as superscript.

Fixed on line 640

Throughout the manuscript, the concept “prion-like” is extensively used. The term has a very substantial currency, but there also is a controversy in the field with three points of view: some authors contend that the term aptly describes the characteristics of tau, Abeta, a-synuclein…propagative aggregates; others, prefer the term “prionoid”, and finally, a third group supports the notion that these are in fact full fledged prions. The main point of contention is the relative infectivity/transmissibility of the various propagative aggregates: several PrPSc strains are clearly infectious and have caused epizootics and epidemics, like mad cow disease, scrapie or Kuru, while others, tau aggregates for example, can be transmitted between individuals under very extreme conditions. The Authors might want to cite Eraña H., Prion. 2019 Jan;13(1):41-45. For a very complete and up to date discussion of this controversy.

The mentioned reference was added (reference 235) along with some accompanying text in lines 687-692.  “We utilize the term prion-like to specifically describe the attributes of templating, strain, and cell to cell transmission shared by several amyloidogenic proteins. However, we also recognize that there is controversy in the field regarding the use of the term “prion-like” vs other terms such as “prionoid” or “Prions” and we direct readers to more targeted review on the subject [235].”

Reviewer 4 Report

The review article gives a good overview about template missfolding, spreading of the misfolding in a non-cell autonomous manner and strains for tauopathies. They summarize these properties of tau under the term “prion-like”, which is an emerging paradigm in the field of proteinopathies. The term prion refers, as reviewed by the authors, to “Proeinacesous Infections Particels” and is a concept that protein only can by an infectious agent transmitting a disease, in the absence of nucleic acid. The difference of prions to other proteinopathies like tau, is that they have an epidemic potential like demonstrated by Kuru and nvCJD. Therefore, prions are pathogens like bacteria and viruses. This difference is also important because the term “prion-like” could imply similar safety measurements like for prions. Therefore, the authors should add a section describing also these differences in contrast to prion disease. In addition, they should define their definition of “prion-like” already in the introduction.

Author Response

We thank the reviewer for taking the time to read our manuscript and offer their constructive feedback.

-We recognize the reviewer's concerns with the term prion-like and clearly defining its distinction from prions from the standpoint of biosafety and public perception. We felt that this is quite a complex topic that we could not adequately discuss with an additional section. We have adjusted the text to direct readers to a more targeted review on the topic.

Altered introductory sentence at lines 567-571:

Tauopathies are considered to be prion-like diseases. The prion-like disease concept centers on appreciating certain overlapping characteristics between prion diseases and various progressive neurodegenerative diseases whose etiologies are propelled by protein misfolding.    

Altered paragraph at lines 667-677:

Initial evidence suggests that other amyloidogenic proteins linked to neurodegeneration might share the transmissible properties ascribed to prions and might thus be considered “prion-like”. We utilize the term prion-like to specifically describe the attributes of templating, strain, and cell to cell transmission shared by several amyloidogenic proteins. However, we also recognize that there is controversy in the field regarding the use of the term “prion-like” vs other terms such as “prionoid” or “Prions” and we direct readers to more targeted review on the subject [235]. The prion-like hypothesis has spurred a transformation of the research landscape for diseases such Alzheimer’s, Parkinson’s, and Huntington’s [59,236-238]. In the following sections, we provide an overview of data examining the prion-like properties of tau. 

-Regarding the reviewer's request to add a definition of prion-like to the introduction preamble: We feel that since we spend a subsection (3.1) introducing the "prion-like” that this was not necessary to pre-define within the short introduction paragraph.

This manuscript is a resubmission of an earlier submission. The following is a list of the peer review reports and author responses from that submission.

Round 1

Reviewer 1 Report

In the review by Alyenbaawi et al., the authors discuss the current data supporting the prion-like behavior of misfolded tau in a group of neurodegenerative disorders called tauopathies. While there are several arguments in the review that are well-developed, there are a number of areas that require additional effort before the manuscript should be published. Those items are discussed in detail below.

  1. The title focuses on TBI, but the section on TBI is underdeveloped and reads as an add-on to a review that is more focused on the prion-like behavior of tau. The authors should consider refocusing the paper to exclude the section on TBI and better develop the other sections of the manuscript.
  2. On page 3, the second paragraph discusses the higher affinity of 4R tau for microtubules than 3R tau. The authors should include the work from Eva Nogales’ lab reporting the cryo-EM structure of tau bound to microtubules to explain why this is the case.
  3. Table 1 – text alignment is off in several boxes and the lines are off on the AD row. Would also be helpful to include disease abbreviations under the disease name give there is space ample to do so.
  4. On page 6, the title of section 2.4 includes ‘toxic structures’ but the authors do not discuss toxicity in the section. The authors should either remove this from the subheading or incorporate discussion about toxicity data.
  5. On page 6, the authors begin a discussion on the role of PTMs in tau aggregation. The authors take the stance that PTMs are a driving factor in tau misfolding, but there is also substantial data suggesting PTMs occur after tau aggregation occurs. The debate between cause vs consequence should be better balanced. Discussion of PTMs on page 10 should also include data showing tau phosphorylation within the RD regulates the affinity of tau for microtubules, contributing to its normal function in the neuron.
  6. On page 7, the authors reference the cryo-EM structures of tau in AD, PiD, and CTE. They should also include the CBD structure.
  7. In Figure 2, the authors show tau aggregates as an anti-parallel beta sheet. All cryo-EM structures from patient-derived material show parallel beta sheets. This should be corrected in the figure.
  8. The section on tau function that starts on the bottom of page 9 should come earlier in the manuscript (perhaps in the first subsection).
  9. The manuscript would benefit from heavy editing. The writing quality is variable. While many areas are concise and straight-forward, there are unfortunately several sentences that are poorly worded, which makes it difficult to understand the point(s) the authors are trying to make. Additionally, these sentences are often very long. The authors fluctuate between verb tense throughout the manuscript, and often in the middle of a sentence, making it difficult to follow. There are frequently words that are pluralized when they should be singular or singular when they should be plural. This combined with incorrect comma usage adds to the confusion with some of the writing.
  10. On page 11, the paragraph that starts on line 348 should be moved earlier in the subsection (before the discussion on impaired clearance).
  11. The authors discuss data pertaining to tau oligomers. It would be very useful to define an oligomer since this word is used to mean a number of things within the literature. (This may need to be done on an experiment-to-experiment basis.)
  12. On page 12, the authors contribute variability in data reported in the literature to differences in oligomers used for studies. It would be useful to consider the contributions that differences in tau conformation contribute given the structural differences between recombinant fibrils and patient-derived material. This is briefly mentioned on page 15, but warrants additional discussion. PrPSc strains exhibit unique biological properties, which might explain much of the discrepancies between studies using human versus recombinant aggregates discussed in the review.
  13. The explanation of prion diseases and the prion templating process on page 12 would benefit from some additional detail. It would also be useful to include examples of prion diseases. Moreover, the subsection would benefit from linking prion diseases with the properties discussed in the following paragraphs. As the manuscript currently reads, the authors must often assume those properties are consistent with prion diseases.
  14. The discussion on page 13 about cell-autonomous mechanisms of disease and selective vulnerability could also be explained by the strain hypothesis. The manuscript would benefit from including a discussion on strains here.
  15. On page 16, the authors talk about the trimer and monomer data reported in the two Mirbaha papers. It would be useful to discuss the biophysics of strains being encoded by monomers and trimers, and how stable either of the two structures are in a flat conformation.
  16. On page 17, the authors mention the cell model reported in Sanders et al. The model uses YFP as a fluorescent tag, not GFP.
  17. On page 18, the authors discuss the structural differences between tau in AD, PiD, CTE, and CBD. Their discussion would be improved by including the differences in isoforms found in the structures and how the isoform incorporation contributes to strains. Additionally, recent work from Virginia Lee’s lab supports the isoform-specific recruitment of tau in different strains in mouse studies (though the title is misleading on this).
  18. Figure 3 does not add anything to the manuscript and should be removed.
  19. A higher resolution image of Figure 4 is needed. Additionally, the authors should edit the legend. For example, SNARE is misspelled as SNEAR and MVB is MBV. It would help to define abbreviations in the legends, as well (or put abbreviations used in the figure in the main text so it is easier for the reader to understand).
  20. In some places the authors capitalize tau and in others it is lower case. Spelling of the protein name should be consistent throughout the manuscript.
  21. The discussion on tau uptake on page 23 should include recent work from Ken Kosik’s lab.
  22. On page 23, the authors discuss differences between the tau species used in uptake studies. For the Holmes paper, they mention the fragment of tau used. They should do the same for the neuronal studies. It is likely that differences between FL and RD tau will contribute to experimental differences.
  23. On page 24, in the sentence that starts on line 879, the authors state that human P301L tau demonstrates that trans-synaptic spread of tau via synapses occurs before disease-associated synaptic loss. This appears to be an incomplete statement since mutant tau itself does not show this.
  24. On page 24, the paragraph that starts on line 883 should either be expanded to explain specifics of studies referenced or incorporated into the previous paragraph if there are no major differences in methods or findings.
  25. The discussion on sleep disruption would benefit from including data reported by Sigrid Veasey’s lab looking at sleep disruption effects on the P301S tau mouse model.
  26. The section on sleep disruption also suggests that shift workers should show an increase in the prevalence of tauopathies. Are the authors aware of any epidemiological studies looking at this?
  27. The section on TBI and CTE feels out of place at the end of the review. If the authors do decide to keep this section, they should address the following issues:
    1. How would the authors propose to conduct the prospective studies suggested at the end of page 25/beginning of page 26?
    2. It is unclear what is added by inclusion of subsection 4.2. This should tie in better with the rest of the section.
    3. In line 1002, NTF should be NFTs and in line 1003, blood vesicles should be blood vessels. Additionally, it would be useful to explain why these particular pathological hallmarks are seen in CTE patients.
    4. In addition to the 1973 paper, the authors should reference the 1927 Martland paper first describing punch drunk syndrome.
    5. Include a discussion on the cryo-EM structure of CTE, including similarities to the AD conformation and thoughts on the potential co-factor (or non-protein density). This is particularly needed given the underdeveloped explanation of what is already known about the tau strain in CTE patients on page 29.
  28. The authors often misspell conformation as confirmation.
  29. In the conclusion, the authors refute the arguments they have made throughout the article by saying there is little evidence to support the prion-like hypothesis for tau. It is unclear which side of this argument they are on.
  30. The authors suggest that a lack of knowledge in the field may be due to inadequate models. They should provide suggestions for how to develop or establish better models.

Reviewer 2 Report

The present manuscript is an excellent review of the tau prion-like properties and tau involvement in neurodegeneration. The only request to increase the quality of the paper should include some more references like the ones listed below:
line 196:
proline-rich domain consists of several PXXP motifs that serve as binding sites for
195 proline-directed kinases such as Fyn [65]. GSK3Beta (Hooper et al., J Neurochem 2008) cdk5 (Kimura et al., Frontiers Mol Neurosci 2014)
Line 205 add Iqbal et al Nat Rev Neurol 2016
In the paragraph starting line 224 it should be included that there is evidence in vitro that tau self assembly could be induced by phosphorylation (Alonso et al., PNAS 2001) and truncation (Kovacech& Novak Current Alzheimer Research 2011)
On line 285 it should cite Baquero et al., Fronteirs in Mol Neurosci 2019.
On line 299 it should add Iqbal et al., BBActa 2005.
In section 3.1 it should be included that the first report that showed that tau from AD can induce tau self assembly is the work of Alonso et al Nat Med 1996
Line 705 it says in vivo validation in vivo, reword
Line 748 it should be added that monomeric tau can also be uptaken ref 262 and Morozova et al., Frontiers in Cell Neurosci 2019

Reviewer 3 Report

Even though there are currently many reviews about the “prion-like” capabilities of tau, this manuscript has potential to be useful to contribute to our understanding in the field of TBI and CTE because few have previously tried to evaluating tauopathy progression in these scenarios.

The review is well written, and it does a good job at summarizing the current literature. However, I would have liked to see a bit some paragraphs discussing the relevance of host tau in addition the source of tau inoculums in seeding and spreading (e.g. Ferrer et al. 2020; Brain pathology) or relevance of phospho-tau in cellular internalization (e.g. Wauters et al. 2016, Biomolecules).

Furthermore, I would advise that the Authors must revise some minor points prior to publication:

-Table 1 format needs to be improved.

-Figure 3 should be smaller.

-Figure legends must be revised again to homogenize the use of the bold letter as well as the indications A, B ... in the panel.

-Line 50: The bar between “tissues” and “or regions” must be removed.

-Symbols for genes (e. g. MAPT) must be italicized (lines 63, 102, 128….)

-The terms ‘in vivo’ and ‘in vitro’ must be written in italics (lines 91, 324, 351…)

-The use of the term “vs.” must be homogenized, with or without an ending period, in roman or italics. See for instance lines 106 and 115.

-Take another look at the sentence starting on line 123 and ending on line 127, as it seems somewhat jumbled to me.

-Line 250: Lack of space between reference 7 and “There”.

-Line 265: Lack of space between “motifs” and reference 93.

-Ending period is missing in some sentences (lines: 280, 335, 1010).

-Lines 331-333: Reference is missing.

-Line 354: “However” must start in capital letters.

-Line 369: Write “tauopathy” instead “Tauopathy”.

-Line 393: Write “tau” instead “Tau”.

-Line 413: Lack of space between “AD” and reference 177.

-Line 477: Extra space (before and after “-α“).

-Line 504: Change “wildtype” to “wild-type”.

-Line 525: Extra ending period.

-Line 591: Lack of space between “(Table 1)” and references 33, 117.

-Line 611: Check the format before “Furthermore”.

-Line 669: Write “tau” instead “Tau”.

-Line 692: “The” must start in capital letters.

-Line 705: Revise double use of “in vivo” in the sentence.

-Line 715: Lack of space between “(…, 2)” and “direct”.

-Line 767: Lack of space between “IL1β” and reference 244.

-Line 862: “Factors” must start in capital letters.

-Take another look at the sentence starting on line 899 and ending on line 901, as it seems somewhat jumbled to me.

-Line 924: Delete “Traumatic brain injury” as this is not the first mention.

-Line 1007: Write “tau” instead “Tau”.

-Line 1032: The abbreviation for human tau (htau) must be indicated in the first mention.